Corrected: Author correction

# Genome-wide mapping of plasma protein QTLs identifies putatively causal genes and pathways for cardiovascular disease

Chen Yao ⃝ iD et al.[#]

Identifying genetic variants associated with circulating protein concentrations (protein quantitative trait loci; pQTLs) and integrating them with variants from genome-wide association studies (GWAS) may illuminate the proteome's causal role in disease and bridge a knowledge gap regarding SNP-disease associations. We provide the results of GWAS of 71 high-value cardiovascular disease proteins in 6861 Framingham Heart Study participants and independent external replication. We report the mapping of over 16,000 pQTL variants and their functional relevance. We provide an integrated plasma protein-QTL database. Thirteen proteins harbor pQTL variants that match coronary disease-risk variants from GWAS or test causal for coronary disease by Mendelian randomization. Eight of these proteins predict new-onset cardiovascular disease events in Framingham participants. We demonstrate that identifying pQTLs, integrating them with GWAS results, employing Mendelian randomization, and prospectively testing protein-trait associations holds potential for elucidating causal genes, proteins, and pathways for cardiovascular disease and may identify targets for its prevention and treatment.

Considerable progress has been made in identifying the genetic underpinnings of coronary heart disease (CHD)[1–3], which remains the leading cause of death worldwide[4]. Proteins, which are the functional products encoded by the genome, serve critical roles in biological processes involved in health and disease, and constitute effective drug targets. While numerous proteins have been reported to be associated with CHD, it is often difficult to establish whether they are causally related to CHD risk or represent downstream markers of CHD-related processes. Therefore, identifying genetic variants associated with circulating protein levels (protein quantitative trait loci; pQTLs), characterizing pQTL variants that also are associated with CHD in genome-wide association studies (GWAS), and inferring causality can provide insights into the roles of genetic variants, genes, and proteins in the pathogenesis of CHD. To date, most pQTL studies have been limited by small sample sizes, lack of integration of pQTL variants with disease-associated SNPs from GWAS, no causal testing, and cross-sectional designs that prevented longitudinal analyses of informative protein-trait associations[5–14].

To address this knowledge gap, we conducted a multistage study (Fig. 1), consisting first of GWAS of high-value plasma proteins associated with cardiovascular disease (CVD) that were measured in Framingham Heart Study (FHS) participants. These GWAS results were then externally replicated in participants from the INTERVAL[15] and Cooperative Health Research in the Region of Augsburg (KORA) studies[11]. Third, we integrated pQTL variants with genetic variants from the CARDIoGRAMplusC4D consortium databases[1–3] and employed Mendelian randomization (MR)[16] to reveal proteins with potentially causal effects on CHD. Last, we tested proteins for associations with new-onset CVD events in FHS participants with long-term follow-up. We demonstrate that our strategy of protein GWAS followed by causal testing and prospective association of proteins with CVD outcomes can identify putatively causal genes, proteins, and pathways for CVD and may highlight targets for its prevention and treatment.

## Results

**Discovery set.** Seventy-one proteins, selected a priori based on prior evidence of association with CVD, were measured in 7333 FHS participants (Supplementary Data 1). The sample size available for GWAS of the 71 proteins was up to 6861 participants (mean age 50 years, 53% women); clinical characteristics of the discovery sample are summarized in Supplementary Data 2.

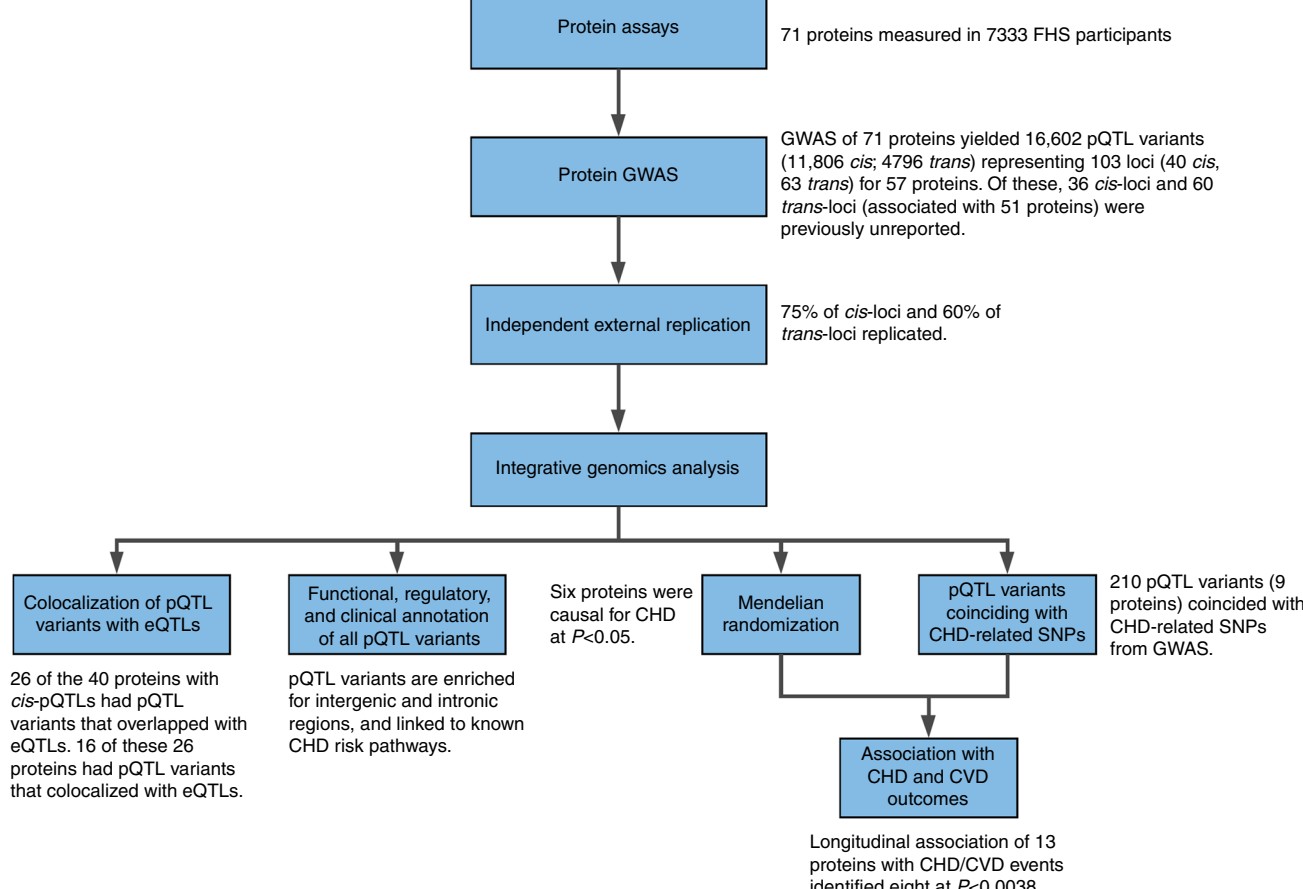

**Fig. 1** Flowchart of Study Design. (1) Selection and measurement of 71 high-value plasma proteins for atherosclerotic CVD via multiplex immunoassays in 7333 FHS participants, (2) GWAS of the 71 proteins in 6861 FHS participants to identify genome-wide significant pQTL variants, (3) independent external replication of sentinel pQTLs in INTERVAL, KORA, and previous GWAS, (4) colocalization and functional enrichment analyses of the identified pQTL variants, (5) integrated analysis of pQTL variants that coincide with CHD SNPs from GWAS, (6) identification of causal proteins for CHD using Mendelian randomization, (7) association analyses of proteins from steps 5 and 6 with risk for new-onset CHD/CVD events in 3520 FHS participants 50 years of age or older with available long-term follow-up. *CHD* coronary heart disease, *CVD* cardiovascular disease, *FHS* Framingham Heart Study, *GWAS* genome-wide association study, *pQTL* protein quantitative trait locus, *SNP* single-nucleotide polymorphism

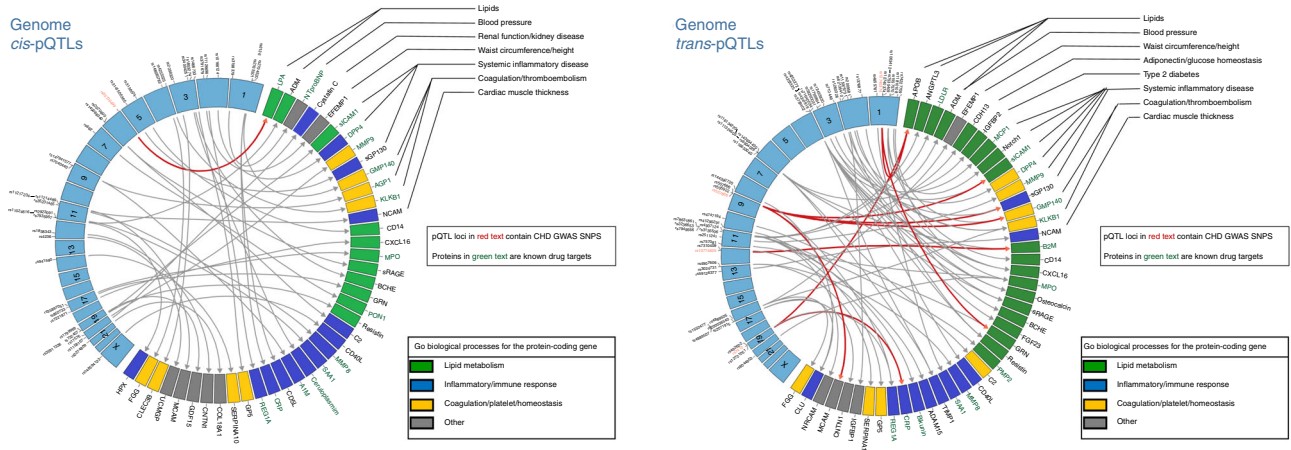

**Fig. 2** Sentinel *cis*- and *trans*-pQTLs and their associated proteins. Circos plots of sentinel *cis*- (left panel) and *trans*-pQTL variants (right panel). Sentinel pQTL variants are listed in order of chromosomal locations (blue boxes in the left semicircle). pQTL variants previously identified in GWAS to be associated with CHD appear in red text. Proteins with genome-wide significant pQTLs are listed in the right semicircle. The following two conditions are summarized for each protein: (1) The corresponding protein-coding gene is a known drug target (green text). (2) GO biological processes for the protein-coding gene (green box denotes lipid metabolism pathways, blue box denotes inflammatory/immune response pathways, yellow box denotes coagulation/platelet/hemostasis pathways, and gray box denotes other pathways not included in the three most common, previously listed pathways). A single primary GO process was chosen when the protein-coding gene was included in multiple pathways. *CHD* coronary heart disease, *GO* Gene Ontology, *GWAS* genome-wide association study, *pQTL* protein quantitative trait locus, *SNP* single-nucleotide polymorphism

**pQTL mapping**. Using Bonferroni correction for multiple testing, we identified 16,602 pQTL variants (with Reference SNP cluster IDs) associated with 57 proteins (Supplementary Data 3), including 11,806 *cis*-pQTL variants (at $P < 1.25E{-}7$, linear regression model unless otherwise stated) for 40 proteins and 4796 *trans*-pQTL variants (at $P < 7.04E{-}10$) for 44 proteins; 27 proteins had both *cis*- and *trans*-pQTL variants. Our study had 80% power to detect a *cis*- or *trans*-pQTL variant that explained ≥0.6% of variance in protein levels (Supplementary Data 4). At each pQTL locus, the SNP with the lowest $P$ value was considered to be the sentinel pQTL variant at that locus. Pruning of the pQTL variants (linkage disequilibrium (LD) $r^2 < 0.1$) yielded 372 non-redundant variants (Supplementary Data 5) representing 103 loci (Supplementary Data 6) consisting of 40 sentinel *cis*-pQTL variants (Fig. 2 left panel) and 63 sentinel *trans*-pQTL variants (Fig. 2 right panel). Among the 16,602 pQTL variants, 341 were coding variants associated with 19 proteins (Supplementary Data 7) and 33 were rare variants (minor allele frequency < 1% genotyped on Exome Chip) associated with 17 proteins (Supplementary Data 8). In addition, 1689 insertion/deletion polymorphisms were identified for 55 proteins (Supplementary Data 9).

The effect sizes and the proportion of inter-individual variation explained by some pQTL variants were large. For example, *cis*-pQTL variant rs941590, a missense SNP previously reported to be associated with family history of venous thrombosis[17], explained 32% of the inter-individual variation in SERPINA10 levels (Supplementary Fig. 1). Three proteins (PON1, GRN, and LPA) had pQTL variants that explained 10–30% of the variation in circulating protein levels. Minor allele frequency was inversely correlated with effect size (P < 0.05, Pearson correlation; Fig. 3), and *cis*-pQTL variants and coding variants tended to have larger effect sizes and explained a greater proportion of the variation in protein levels than did *trans*-pQTL variants and non-coding variants, respectively (P < 0.05, Student's *t*-test; Fig. 3).

**External replication**. Among our 103 sentinel pQTL variants linked to 57 proteins, 96 sentinel pQTL variants(36 *cis*- and 60 *trans*-pQTL variants) associated with 51 proteins were not

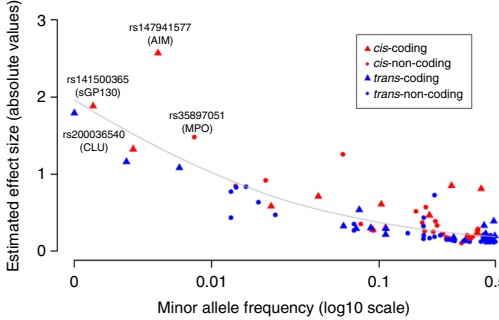

**Fig. 3** pQTL variant minor allele frequency vs. effect size. Minor allele frequency of sentinel pQTL variants (*X*-axis) vs. effect size of variants on proteins. The average absolute estimated effect size (per standard deviation per allele) is significantly different (by the unequal variance *t*-test) between coding and non-coding pQTL variants (0.56 versus 0.31, $P = 0.02$), and also significantly different between *cis* and *trans*-pQTL variants (0.53 vs. 0.30, $P = 0.017$). *pQTL* protein quantitative trait locus

previously reported in GWAS. We attempted to replicate all 103 sentinel pQTL variants in the INTERVAL[15] ($N = 3301$) and the KORA[11] studies ($N = 997$). Among our 57 proteins linked to 103 sentinel pQTL variants, 45 proteins (associated with 32 sentinel *cis*-pQTL variants and 56 sentinel *trans*-pQTL variants) were independently measured in the INTERVAL study. Of the 32 sentinel *cis*-pQTL variant-protein pairs (for 32 proteins) from the FHS, 23 (72%) replicated in INTERVAL at $P < 5.7E{-}4$ (alpha level of 0.05 after Bonferroni correction for 88 tests; 0.05/88). Of the 56 sentinel *trans*-pQTL variant-protein pairs (for 37 proteins) from the FHS, 33 (59%) *trans*-pQTL variant-protein pairs (for 22 proteins) replicated in INTERVAL (P < 5.7E−4; Supplementary Data 10). One additional *trans*-pQTL variant-protein pair replicated in KORA. Four proteins (associated with four *cis*-pQTL variants and one *trans*-pQTL variant) that were not measured or did not replicate in INTERVAL or KORA replicated based on prior GWAS evidence (Supplementary Data 10). The remaining 10 proteins without any available external source of replication were associated with six *cis*- and six *trans*-pQTL

variants (Supplementary Data 10). In total, 27 (75%) sentinel *cis*-pQTL variants and 34 (60%) sentinel *trans*-pQTL variants replicated (at Bonferroni-corrected $P < 0.05$) with 100% consistent direction of effect compared with the FHS discovery results.

**Resampling analysis**. Based on 1000 resamplings of 3300 FHS participants in the discovery sample, 80 pQTL-protein associations (31 *cis* and 49 *trans*) yielded $P < 5.7E-4$ (alpha level of 0.05 after Bonferroni correction for multiple testing; 0.05/88) in ≥80% of samplings and thus were considered likely to replicate in a GWAS sample size of 3300 individuals from INTERVAL (Supplementary Data 10). Among the 80 pQTL-protein associations that were considered likely to replicate, 54 (68%) replicated in INTERVAL. The discrepancy between predicted and observed replication may be due in part to proteomic platform differences between the discovery and replication studies.

**pQTL functional and regulatory annotations**. We explored the function annotation of each protein. Some of the genes coding for CHD-related proteins are linked to known CHD risk pathways via previous GWAS of lipids (APOB, LPA, ANGPTL3), coagulation pathways (GMP140), and systemic inflammation (sGP130, sICAM1) as shown in Fig. 2. Many of the proteins that share genetic underpinnings with CHD are known drug targets (DrugBank database)[18], or are currently under development as such (e.g., ANGPTL3, LPA, sICAM1, and GMP140). Several proteins with pQTLs linked to CHD, however, are not known drug targets, particularly those from genetic loci not previously linked to CHD risk pathways (e.g., BCHE, CXCL16, MCAM, and sRAGE).

**Colocalization of pQTLs and eQTLs**. Among the 372 non-redundant pQTL variants, we identified 190 unique variants (associated with 53 proteins) that were also eQTL variants (genetic variants associated with whole blood gene expression levels in FHS participants[19]) at FDR < 0.05. These 190 eQTL variants consisted of 188 *cis*-eQTL variants and 27 *trans*-eQTL variants (Supplementary Data 11), suggesting that a substantial number of causal eQTL variants may also be causal pQTLs. To test this hypothesis, we conducted a Bayesian test of

colocalization of *cis*-pQTL variants using the coloc package in R for genes within 1 Mb region (upstream or downstream) of each sentinel *cis*-pQTL variant[20] (see Methods). Among the 40 sentinel *cis*-pQTL variants, 26 were associated with the expression of genes residing within 1 Mb (FDR < 0.05), and these 26 unique lead SNP-transcript-protein pairs were tested for colocalization. Using all SNPs shared by transcripts and proteins, we conducted a colocalization test for each protein to determine the probability that the two association signals were due to the same causal variant. The prior probabilities for a SNP being associated with gene expression only (p1), protein level only (p2), or with both traits (p12) were based on the number of eSNPs and pQTL variants observed in our data (see Methods). The value for p12 was set to 75%, i.e., the probability that a causal eSNP is a causal pQTL variant. For 16 out of 26 proteins that were associated with both *cis*-pQTL variants and eQTL variants, we observed a probability > 75% that the pQTL variants colocalized with the eQTL variants (Supplementary Data 12). We observed similar colocalization results by applying default *P* values (probability values, p1 = 1E−4, p2 = 1E−4, p12 = 1E−5) in the coloc package assuming that 1 in 10,000 SNPs is causal for either trait.

**Integrating pQTL variants with CHD-associated SNPs**. We integrated our pQTL variants with 2213 CHD-related SNPs from the CARDIoGRAMplusC4D Consortium GWAS[1–3]. A total of 210 pQTL variants (16 non-redundant variants at LD $r^2 < 0.1$ representing nine proteins; Supplementary Data 13) exactly matched SNPs associated with CHD from prior GWAS. Table 1 displays the sentinel pQTL variants that coincided with CHD-related GWAS SNPs and the corresponding protein at each genetic locus. The proteins with pQTL variants coinciding with CHD-associated SNPs included LPA, APOB, B2M, CRP, GMP140, GRN, MCAM, sGP130, and sICAM1. It is important to note, however, that these results do not indicate a causal relationship between the pQTL-associated protein and CHD.

We found the *ABO* locus to have links to CHD through four circulating proteins (GMP140, MCAM, sGP130, sICAM1). ABO blood type has previously been linked to CVD risk in the FHS[21], and additional reports have linked the *ABO* locus to CVD via coagulation pathway effects[22,23]. The proteins related to the *ABO* locus that were identified in our study are involved in

**Table 1 Proteins with pQTL variants that coincide with coronary heart disease-associated SNPs from Genome-wide Association Studies**

| Measured protein | pQTL variant[a] | Location | pQTL-annotated gene | pQTL-protein association | | | | pQTL-CHD association | |
|---|---|---|---|---|---|---|---|---|---|
| | | | | Effect allele | EAF | Beta | *P* value | Beta | *P* value[c] |
| APOB | rs7412 | 19:45412079 | *APOE* | T | 0.076 | −0.54 | 9.62E−53 | −0.14 | 8.17E−11 |
| APOB | rs12740374 | 1:109817590 | *CELSR2* | T | 0.215 | −0.18 | 4.95E−16 | −0.11 | 4.63E−23 |
| B2M | rs10774625 | 12:111910219 | *ATXN2* | A | 0.497 | 0.12 | 8.66E−11 | 0.067 | 2.69E−10 |
| CRP | rs12721051 | 19:45422160 | *APOC1* | C | 0.851 | 0.23 | 2.39E−17 | −0.091 | 1.98E−10 |
| GMP140 | rs507666 | 9: 136149399 | *ABO* | A | 0.818 | −0.43 | 1.11E−72 | 0.080 | 1.64E−11 |
| GRN | rs12740374 | 1:109817590 | *CELSR2* | G | 0.215 | 0.75 | 2.72E−268 | 0.11 | 4.63E−23 |
| LPA | rs55730499[b] | 6:161005610 | *LPA* | C | 0.939 | −1.25 | 3.77E−167 | −0.32 | 4.66E−09 |
| MCAM | rs507666 | 9: 136149399 | *ABO* | A | 0.185 | −0.16 | 2.45E−11 | 0.079 | 1.64E−11 |
| sGP130 | rs507666 | 9: 136149399 | *ABO* | A | 0.186 | −0.21 | 1.68E−18 | 0.079 | 1.64E−11 |
| sICAM1 | rs507666 | 9: 136149399 | *ABO* | A | 0.185 | −0.32 | 4.39E−42 | 0.079 | 1.64E−11 |

*EAF* effect allele frequency, *CHD* coronary heart disease, *pQTL* protein quantitative trait locus
[a]For proteins with multiple pQTL variants that coincide with coronary heart disease GWAS SNPs, the pQTL variant with the lowest *P* value of association with its corresponding protein level is shown
[b]Indicates *cis*-pQTL. All other pQTLs shown in this table are *trans*-pQTLs
[c]*P* value of associations with coronary heart disease risk in GWAS reported in the CARDIOGRAMplusC4D Consortium

inflammatory pathways such as interleukin-mediated and interferon-mediated signaling (Fig. 2). We hypothesize that the multi-protein association of this locus may be due to the function of the *ABO*-encoded protein as a glycosyltransferase.

***Trans*-pQTLs and CHD**. Of the nine proteins with pQTL variants that precisely matched CHD-associated SNPs, eight had pQTL variants with *trans* effects (54 non-redundant *trans*-pQTL variants in total) and 69% of these *trans*-pQTL variants were also associated with the expression of nearby genes (*cis*-eGenes), i.e., these *trans*-pQTL variants were also *cis*-eQTL variants associated with the expression of nearby *cis*-eGenes. Based on these findings, we hypothesized that *trans*-pQTL variants may regulate circulating protein levels through *cis*-effects on the expression of nearby *cis*-eGenes (Supplementary Fig. 2a). To test this hypothesis, we employed Mendelian randomization (MR)[16] using the expression of all genes within 1 Mb of the *trans*-pQTL locus as the exposure, *cis*-eQTL variants associated with these genes (from the FHS whole-blood gene expression database[19]) as instrumental variables, and circulating protein levels as the outcome. We found that for eight proteins the effects of *trans*-pQTL variants on circulating protein levels were causally regulated by the expression of *cis*-eGenes (Supplementary Data 14). For example, we found decreased *SH2B3* expression to be causal for higher circulating B2M levels. This extends prior knowledge of the associations of *SH2B3*[24] and B2M[25] with hypertension as we demonstrate a unidirectional causal association between *SH2B3* expression and plasma B2M levels, and thus provide plausible evidence for a causal role of the *SH2B3*-B2M axis in hypertension (Supplementary Fig. 2b). To extend these findings to other CHD-related tissues, we applied the same analyses to GTEx[26] whole-blood, liver, and heart eQTLs. For two of the proteins (APOB and GRN), there was additional experimental evidence in support of our results through interrogation of GTEx[27] whole blood eQTLs (Supplementary Data 14). Moreover, we found significant causal effects of *CELSR2/SORT1/PSRC1* on APOB levels (in liver), *CELSR2/SORT1/PSRC1* on GRN levels (in artery), and *ABO* on GMP140 levels (in heart atrial appendage).

**Causal testing**. We applied MR testing using pruned *cis*-pQTL variants (LD $r^2 < 0.1$) as instrumental variables for circulating protein levels in order to identify proteins that were causal for CHD. MR testing was conducted for all 40 proteins with *cis*-pQTLs and causally implicated LPA, BCHE, PON1, MCAM, MPO, and cystatin C ($P < 0.05$; MR test; Supplementary Data 15). Causal testing for LPA and BCHE remained statistically significant after adjusting for multiple testing ($P < 0.05/40$).

**Protein associations with clinical outcomes**. For the 13 proteins with pQTL variants that either coincided with CHD GWAS SNPs (nine proteins) or tested positive by MR (six proteins) at $P < 0.05$, we sought to determine the longitudinal associations of circulating levels of these proteins with (a) major CHD events (recognized myocardial infarction or CHD death; $n = 213$ events) and (b) CVD death (fatal CHD or death due to stroke, peripheral arterial disease, heart failure, or other CVD causes; $n = 199$ events) with a median follow-up of 14.3 years (25th percentile 11.4, 75th percentile 15.2 years) among 3520 FHS participants 50 years of age or older. Eleven of the 13 proteins were nominally associated ($P < 0.05$, linear regression) with incident CHD and/or CVD death (Table 2), and eight proteins remained statistically significant after adjusting for multiple testing ($P < 0.05/13$). Two of the six proteins (PON1 and cystatin C) that tested causal for CHD by MR at $P < 0.05$ were also associated with long-term CHD/CVD outcomes at $P < 0.0038$ (Fig. 4). The protein effect sizes on CHD predicted from MR were directionally consistent with the observed prospective protein-CHD associations in all cases except for PON1 (Fig. 5).

**Table 2 Protein associations with coronary heart disease events and cardiovascular disease death in Framingham Heart Study participants with long-term follow-up**

| Protein | Association with CHD in GWAS, MR, or both | KEGG pathways for which pQTLs of a protein are enriched[b] | Association with CHD events[a] | | Association with CVD death[a] | |
|---|---|---|---|---|---|---|
| | | | Hazards ratio (95% CI) | *P* value[c] | Hazards ratio (95% CI) | *P* value[c] |
| APOB | CHD GWAS | Endocytosis | 1.44 (1.24–1.67) | **1.8E−06** | 1.07 (0.91–1.26) | 0.41 |
| B2M | CHD GWAS | -Type I diabetes mellitus -Antigen processing and presentation -Allograft rejection -Graft vs. host disease -Autoimmune thyroid disease | 1.47 (1.24–1.75) | **9.0E−06** | 1.97 (1.63–2.38) | **2.14E−12** |
| CRP | CHD GWAS | None | 1.40 (1.20–1.62) | **1.41E−05** | 1.43 (1.23–1.68) | **5.78E−06** |
| GMP140 | CHD GWAS | Cell adhesion molecules | 1.23 (1.06–1.42) | 0.0071 | 1.25 (1.06–1.46) | 0.0063 |
| GRN | CHD GWAS | None | 1.14 (0.98–1.32) | 0.081 | 1.29 (1.11–1.51) | **0.0012** |
| LPA | Both | None | 1.09 (0.94–1.26) | 0.24 | 1.09 (0.93–1.27) | 0.29 |
| MCAM | Both | None | 0.88 (0.76–1.03) | 0.10 | 1.11 (0.95–1.30) | 0.19 |
| sGP130 | CHD GWAS | None | 1.02 (0.88–1.18) | 0.80 | 1.34 (1.14–1.56) | **0.00028** |
| sICAM1 | CHD GWAS | Complement/coagulation cascades | 1.23 (1.06–1.42) | 0.0051 | 1.33 (1.14–1.55) | **0.00022** |
| PON1 | MR | None | 0.65 (0.56–0.75) | **1.83E−08** | 0.78 (0.66–0.92) | **0.0035** |
| BCHE | MR | None | 1.09 (0.94–1.26) | 0.25 | 0.82 (0.70–0.96) | 0.013 |
| MPO | MR | -ABC transporters -Type II diabetes mellitus | 1.16 (1.00–1.34) | 0.045 | 1.24 (1.07–1.44) | 0.0047 |
| Cystatin C | MR | None | 1.47 (1.24–1.73) | **4.1E−06** | 1.77 (1.48–2.13) | **5.4E−10** |

Hazards ratios, adjusted for age and sex, are represented per standard deviation increase in inverse-normalized protein level
*CHD* coronary heart disease, *CI* confidence interval, *CVD* cardiovascular disease, *GWAS* genome-wide association study, *KEGG* Kyoto Encyclopedia of Genes and Genomes, *MR* Mendelian randomization
[a]CHD events ($n = 213$) included recognized myocardial infarction or death from CHD, and CVD death ($n = 199$) included fatal CHD or death due to stroke, peripheral arterial disease, heart failure, or other CVD causes occurring during a median follow-up of 14.3 years (25th percentile 11.4, 75th percentile 15.2 years) among 3520 Framingham Heart Study participants age ≥ 50 years
[b]For proteins with pQTL variants that are enriched for more than five KEGG pathways, the top five most significant pathways based on enrichment *P* value are shown
[c]The *P* value threshold for significance ($P < 0.0038$) was determined by the Bonferroni method (0.05/13 proteins tested). Significant *P* values are shown in bold

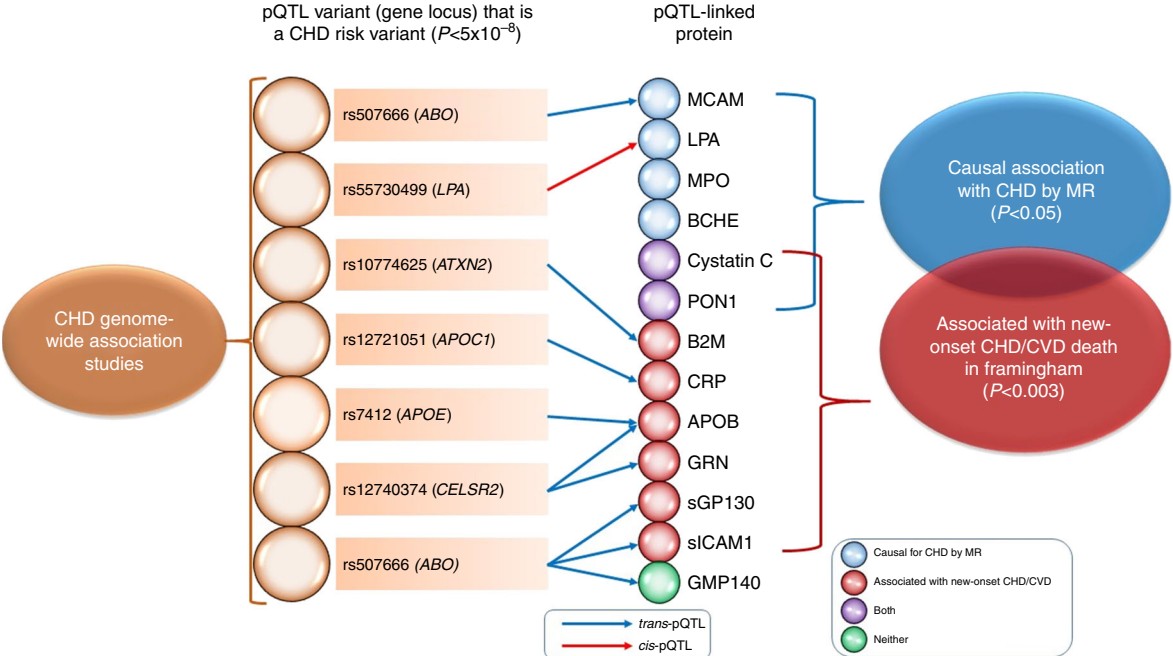

**Fig. 4** pQTL-protein-coronary heart disease network. Network of proteins and significant pQTL variants that are also GWAS risk SNPs for CHD. For proteins with multiple pQTL variants that coincide with CHD GWAS SNPs, the pQTL variant with the lowest *P* value of association with its corresponding protein level is shown. The following two conditions are summarized: (1) Proteins that tested causal for CHD in Mendelian randomization (*P* < 0.05). (2) Proteins associated with new-onset major CHD/CVD events (*P* < 0.0038) in 3520 Framingham Heart Study participants 50 years of age or older with long-term follow-up. Proteins in green fulfill neither condition 1 nor 2; proteins in blue fulfill condition 1; proteins in red fulfill condition 2; proteins in purple fulfill conditions 1 and 2. *CHD* coronary heart disease, *FHS* Framingham Heart Study, *MR* Mendelian randomization, *pQTL* protein quantitative trait locus

**Proteins and pathways implicated in CHD.** Six proteins were implicated by MR analyses as nominally causal for CHD (Supplementary Data 15). LPA, which interferes with the fibrinolytic cascade[28], has been previously demonstrated to be causal for CHD[29], and served as a positive control. BCHE has previously been reported to be inversely associated with long-term CVD mortality[30], and several polymorphisms within *BCHE* have been reported[31] to be associated with CHD risk factors. rs1803274, the sentinel *cis*-pQTL variant for BCHE (A539T) (Supplementary Data 6), is associated with decreased BCHE circulating levels and enzymatic activity[32], and has been shown to predict early-onset CHD[31]. Our protein-trait analyses similarly demonstrated inverse associations between plasma BCHE and long-term cardiovascular outcomes, and MR analyses revealed lower BCHE to be causal for CHD.

Four additional proteins were nominally causal for CHD by MR. PON1 exhibits cardioprotective effects through prevention of LDL oxidation[33], and overexpression of *PON1* in mice inhibits the development of atherosclerosis[34]. Our protein-trait analyses similarly demonstrated an inverse association between PON1 levels and long-term cardiovascular outcomes, but MR revealed that higher PON1 levels are causal for CHD (Table 2). We hypothesize that this directional discordance may reflect down-regulation of *PON1* expression in the setting of CHD. MPO, which promotes formation of atherosclerotic lesions by enhancing APOB oxidation within circulating LDL particles[35], was positively associated with incident cardiovascular outcomes in our protein-trait and MR analyses. Cystatin C, a pro-atherosclerotic[36] cysteine proteinase cathepsin inhibitor and well-characterized biomarker of CHD risk[37], was also positively associated with CVD events in our protein-trait and MR analyses.

MCAM, or CD146, was causally associated with CHD risk in an inverse manner by MR. This is directionally consistent with prior animal studies of limb ischemia, which have shown that injection of sCD146 into the circulation decreased fibrosis and inflammation and increased local perfusion[38].

Several proteins lacked *cis*-pQTLs and therefore were unavailable for MR analysis. Six of these proteins had pQTL variants that perfectly matched CHD SNPs from GWAS and were associated with CHD/CVD outcomes in FHS participants with long-term follow-up: GRN, sGP130, sICAM1, APOB, B2M, and CRP. GRN has previously been implicated in atherosclerosis progression and incident MI[39]. Its precursor, progranulin, has been shown to bind to SORT1, which contains a *trans*-pQTL variant for GRN that is also associated with CHD[40]. sGP130 levels have been shown to positively correlate with long-term CVD mortality[41], perhaps via pathways related to hypertension and vascular remodeling[42]. B2M, an essential component of the major histocompatibility complex I[43], is associated with hypertension[44], atherosclerosis, and CVD[45]. Finally, circulating APOB, an LDL particle ligand, is a well-characterized biomarker of CVD risk[46].

**Molecular QTL browser.** Our full pQTL results are available for download through ftp://ftp.ncbi.nlm.nih.gov/eqtl/original_submissions/FHS_pQTLs/ and the searchable results are accessible through the NCBI Molecular QTL Browser (https://preview.ncbi.nlm.nih.gov/gap/eqtl/studies/). The browser links our pQTL results to eQTLs and other molecular resources via a user-friendly interface. Users can browse and filter the results, for example, by *P* value cutoffs. The Molecular QTL Browser also permits users to conduct targeted studies of specific genes based on prior evidence.

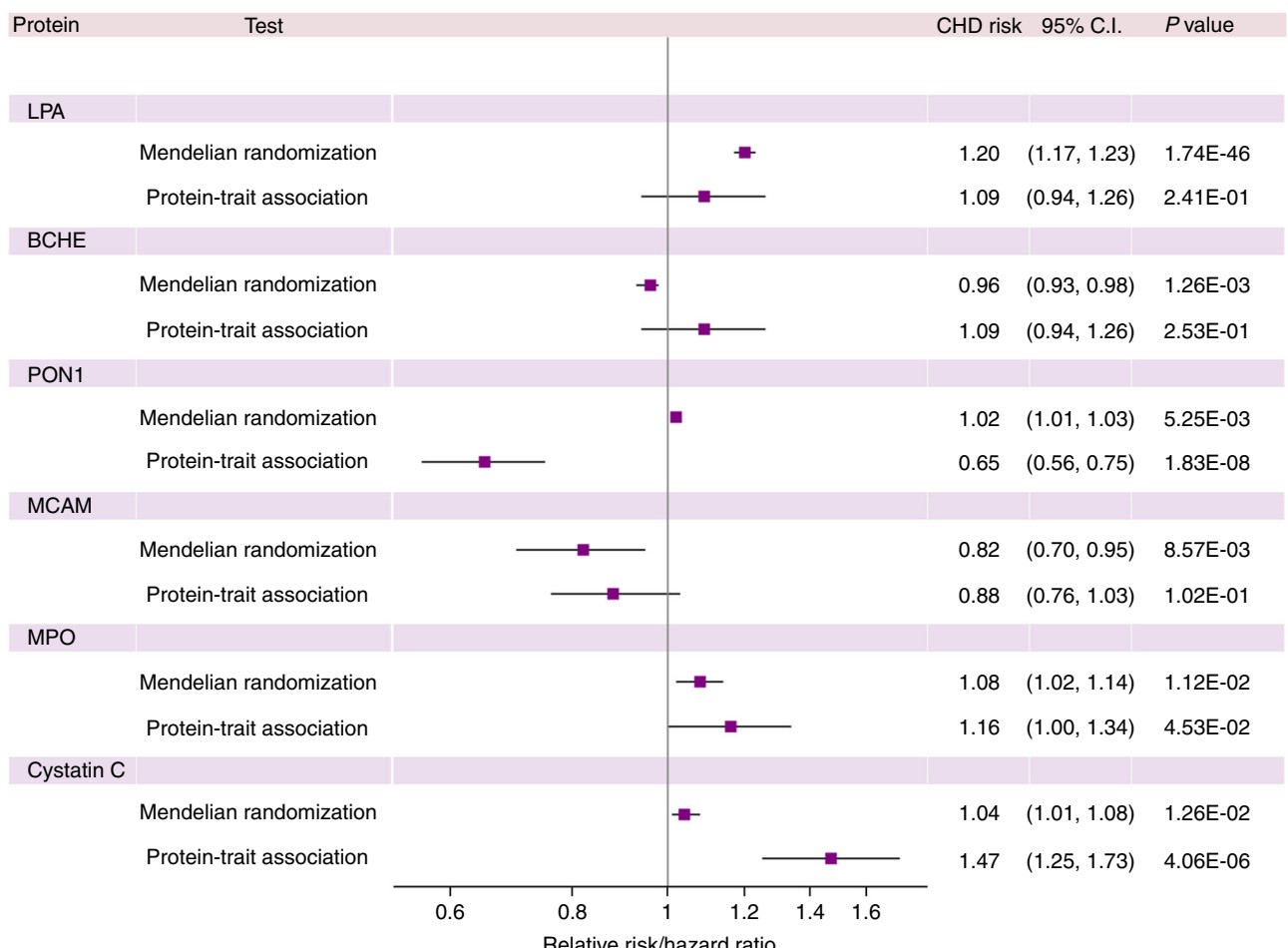

**Fig. 5** Protein effects on coronary heart disease from Mendelian randomization and observed protein-trait associations. A comparison of protein effects on risk of CHD estimated from Mendelian randomization vs. the observed protein-trait associations hazards ratios. *CHD* coronary heart disease, *CI* confidence interval; CHD risk (per standard deviation increase in inverse-normalized protein level)

The integrated data resource enables searches across data sets and filtering by functional annotation and genomic position.

## Discussion

Using a multistage strategy, we discovered over 16,000 pQTL variants associated with 57 out of 71 proteins that were selected a priori as high-value plasma proteins for CVD. Integration of pQTL variants with CHD risk variants from GWAS revealed nine proteins with pQTL variants that coincided with CHD risk variants from prior GWAS (Table 1), and MR analyses implicated six proteins as causal for CHD (Supplementary Data 15). Two of the CHD-causal proteins were also associated with new-onset CHD or CVD events in FHS participants with long-term follow-up (Table 2). Thus, by integrating pQTL variants and proteins with long-term follow-up for clinical events (Fig. 4), we demonstrated a comprehensive approach for identifying putatively causal genes, proteins, and pathways involved in CVD and, in doing so, bridged a previous GWAS knowledge gap for genetic variants that lacked known mechanistic links to disease.

We acknowledge several limitations of our study. First, participants were of European ancestry, and therefore the results may not be directly generalizable to populations with different ethnic/racial backgrounds. Second, *cis*-pQTL coding variants alter the amino acid sequence of the coded protein and may impact the quantitative protein assay. Third, while our GWAS sample size was large, our power to detect and test proteins for causality by MR was limited. Finally, protein levels were measured in whole blood and may not accurately reflect tissue-specific patterns of expression. Furthermore, our gene transcript levels and circulating protein levels were not measured at the same point in time, which may limit the power to colocalize protein-coding genes with their corresponding proteins.

We have provided a large and comprehensive compilation of pQTL variants (via the NCBI Molecular QTL Browser) and show that an integrated genomic approach can identify proteins with putatively causal effects on disease risk. While some of our causally implicated proteins act through classic CVD risk factors and known pathways, others likely act through previously unidentified pathways and thus represent novel targets for drug development. Additional human and animal studies are needed to confirm our MR results and elucidate the mechanisms by which such proteins alter CHD risk. Taken together, the pQTLs identified and analyzed in this study elucidate genes, proteins, and pathways related to CHD, and may have profound implications for the treatment and prevention of the leading cause of death worldwide.

## Methods

**Study design**. The study consisted of seven steps (Fig. 1): (1) selection and measurement of 71 plasma proteins associated with atherosclerotic CVD via multiplex immunoassays in 7333 FHS participants, (2) genome-wide association study of the 71 proteins in 6861 FHS participants to identify genome-wide significant pQTL variants, (3) independent external replication of the sentinel pQTL variants in INTERVAL, KORA and other previous GWAS, (4) functional enrichment analyses of the identified pQTL variants, (5) integrative analyses of

pQTL variants coinciding with CHD GWAS SNPs, (6) identification of proteins causal for CHD using Mendelian randomization, (7) association analyses of proteins from steps 5 and 6 with risk for incident CHD and CVD death in 3520 FHS participants aged 50 years or older with available long-term follow-up.

**Discovery study sample**. The FHS, a community-based prospective study of CVD and its risk factors, consists of three generations of participants within families recruited in 1948, 1971, and 2002[47–49]. The study samples for this investigation were collected from 7333 participants from the FHS Offspring (Exam 7; 1998–2001) and Third Generation (Exam 1; 2002–2005) cohorts. The final sample for GWAS consisted of 6861 participants with complete imputed dosage data based on the 1000 Genomes Project reference panel (1000 G build 37 phase 1 v3)[50]. For association analyses using Exome Chip genotypes (see Genotyping for details), the sample size was 6763. Genome-wide analysis of SNPs associated with gene expression levels (eQTLs) was performed on 5257 FHS participants from Offspring Exam 8 and Third Generation Exam 2 in whom both genotype and gene expression data were available[19].

**Replication study sample**. The INTERVAL bioresource is a cohort of ~50,000 whole blood donors recruited from across England between 2012 and 2014[15,51]. After QC, the subset of 3301 participants with imputed genetic data and protein measurements were included in the replication analyses[52]. The KORA F4 study is a prospective population-based cohort study consisting of 3080 participants living in Augsburg, Southern Germany[11,53], of which a total of 1000 participants who also participated in a metabolomic study with follow-up information for age-related diseases composed the replication study population. After excluding participants with missing genotype or protein data ($n = 3$), the final KORA sample included 997 individuals.

**Power calculation**. For power in the discovery stage with $n = 6800$, we assumed an additive genetic model with no gene-gene interactions, and a population mean = 0 and standard deviation = 1 for all rank-normalized protein levels. At $\alpha = 1.25E{-}7$ for *cis*-pQTL variants and $7.04E{-}10$ for *trans*-pQTL variants for a two-sided test, power was estimated for MAF = 0.002, 0.005, 0.01, 0.05, 0.1, 0.2, 0.3, 0.4, and 0.5 with QUANTO[54]. For an estimate of empirical power in the replication stage (sample size of ~3300 in INTERVAL), we performed pQTL analyses with 1000 resamplings of 3300 FHS participants. Half of them were unrelated and the rest were randomly sampled. We counted the number of results with $P < 0.05/n$ in the 1000 resamplings, where $n$ is the number of pQTL variants that were tested for replication in INTERVAL.

**Clinical measures**. All FHS participants underwent periodic clinical examinations with standard protocols[49]. A three-physician panel was formed to perform medical chart review weekly. The review panel jointly assigned CVD diagnoses and causes of death. All suspected CVD events were adjudicated by a physician panel upon review of all available medical evidence including hospital records, personal physician records, and, in the event of out-of-hospital deaths, interviews with next of kin. Recognized myocardial infarction (MI) was diagnosed when two of three of the following conditions were present: prolonged chest discomfort or symptoms of coronary ischemia, elevated biomarkers of myocardial necrosis (e.g., CK-MB or troponin), and the development of new diagnostic Q-waves on the ECG. Fatal CHD events included fatal MI and other deaths due to CHD as the underlying cause in the absence of evidence of recent MI. Fatal CVD events additionally included deaths due to stroke, peripheral arterial disease, heart failure, or other cardiovascular causes.

**Protein quantification**. FHS fasting blood plasma samples were collected and stored at $-80\,^\circ$C. Candidate protein biomarkers were selected a priori based on previous evidence of association with atherosclerotic CVD or its risk factors using the following complementary approaches: (a) comprehensive literature search[55], (b) proteomics discovery via mass spectrometry in the FHS or elsewhere[39,56], and (c) targeting proteins coded by genes identified via gene expression profiling studies[57,58] or GWAS[59] of atherosclerotic CVD and its risk factors (Supplementary Data 1). A total of 85 plasma protein biomarkers were assayed using a modified enzyme-linked immunosorbent assay sandwich method, multiplexed on a Luminex xMAP platform (Luminex, Inc., Austin, TX). All targets were first developed as singleton assays before compatible targets were pooled to create multiplex panels. Standard Luminex assays with previously published methods were used[60,61]. Measurements were calibrated using a seven-point calibration curve (in triplicate) and tested for recovery at both ends of the quantitation scale. The "High" and "Low" spike controls (QC1 and QC2 respectively) were used to calculate intra-assay and inter-assay coefficients of variation (CV) for each protein. A total of 14 proteins had low call-rate (<90%) mainly due to values falling below the lower detection limit that were excluded for the current study. A list of the 71 proteins and their coefficients of variation and selection criteria were shown in Supplementary Data 1.

For the KORA study, plasma levels of 1129 proteins in 977 participants were measured using the SOMAscan platform (SomaLogic Inc., Boulder, Colorado), a multiplexed aptamer-based affinity proteomics platform; 1124 proteins passed

quality control. Protein measurement protocol, normalization of protein values, and data quality are described elsewhere[12]. For the INTERVAL study, plasma levels of 3620 proteins were assayed using an extended version of the SOMAscan platform[52].

**Genotyping**. Genotyping and QC were conducted in the FHS[17]. In short, genome-wide genotyping was conducted using the Affymetrix 500 K mapping arrays, 50 K supplemental Human Gene Focused arrays (Affymetrix, Inc., Santa Clara, CA), and Illumina Human Exome BeadChip v.1.0 (Exome Chip; Illumina, Inc., San Diego, CA). Genotypes from the Affymetrix arrays were used in conjunction with the 1000 G reference panel build 37 phase 1 v3[50] to generate an imputed set of ~30 million variants using MACH[62]. SNPs with imputation quality ratio $r^2 < 0.5$ (the imputation quality ratio is calculated as the ratio of the variances of the observed and estimated allele counts) were excluded, leaving a final set of 8,509,364 SNPs for 1000 genomes-imputed GWAS.

The Exome Chip includes predominantly rare coding variants not covered by previous genotyping arrays[63]. More than 90% of the SNPs included in the Exome Chip are non-synonymous, splice, or stop codon-altering variants. Common variants on the Exome Chip include 5542 SNPs that were selected based on their associations with disease traits reported in the NHGRI GWAS Catalog[1]. Rare variants with MAF < 1 × 10$^{-4}$ were excluded from the analyses.

For INTERVAL, the Affymetrix Axiom UK Biobank Array (Affymetrix, Inc., Santa Clara, CA) was used to assay approximately 830,000 variants at Affymetrix (Santa Clara, California, US), which were imputed to a combined 1000 Genomes/UK10K reference panel. For KORA, the same array was used to genotype 3788 study participants[11,53]. Genotypes were then imputed from the 1000 G reference panel[14] and used to lookup the replication targets.

**Functional annotation of pQTLs**. We used the Functional Mapping and Annotation of GWAS (FUMA; http://fuma.ctglab.nl)[64] database to categorize proteins based on known pathways and conduct functional annotation of pQTLs (regional plot of each pQTL locus, functional categorization of pQTL SNPs, gene mapping, and pathway enrichment analyses).

**Gene expression**. Gene expression profiling was conducted using the Affymetrix Human Exon 1.0 ST GeneChip platform (Affymetrix Inc., Santa Clara, CA), comprised of >5.5 million probes covering expression of 17,873 mRNA transcripts. Gene expression values were normalized and adjusted for three technical covariates (batch, first principal component, and residual of probeset mean values) as described previously[19].

**Coronary heart disease-associated SNPs**. The CARDIoGRAMplusC4D Consortium[1] GWAS of CHD yielded 2213 genome-wide significant SNPs (at $P <$ 5E$-8$) from 1000 G imputation.

**Genome-wide association (pQTL) analyses**. Statistical analyses in the FHS were performed using R software version 3.1.1[65] or SAS software version 9.4. Linear mixed effects models (the "LMEKIN" function of Kinship Package in R) were used to test associations of inverse-rank normalized protein levels with 1000 G or Exome Chip variants using an additive genetic model. A *cis*-pQTL variant was defined as a SNP residing within 1 megabase (Mb) upstream or downstream of the transcription start site of the corresponding protein-coding gene. A SNP located > 1 Mb upstream or downstream of the gene transcript or on a different chromosome from its associated gene was categorized as a *trans*-pQTL variant. We estimated that there were 440,409 potential *cis* SNP-protein pairs in total. Therefore, the Bonferroni-corrected $P$ value for *cis*-pQTL variants was calculated as 0.05/440,409 = 1.25E$-7$. The number of potential *trans* SNP-protein pairs was 8.5 million (SNPs) × 71 (proteins) − 440,409, yielding a Bonferroni-corrected $P$ value for *trans*-pQTLs of 7.04E$-10$. Linkage disequilibrium (LD) was computed as the square of Pearson's correlation ($r^2$) between imputed additive dosages of genotypic variants within the same chromosome across 8481 FHS individuals with genotype data. Independent pQTLs for a given protein were defined as those with LD $r^2 < 0.2$ with other pQTLs at a genomic locus. For a genetic locus with multiple pQTLs in LD (i.e., LD $r^2 > 0.2$), the pQTL with the lowest $P$ value was selected as the sentinel pQTL for that locus.

For KORA, linear regression models were performed on the follow-up SNPs using R version 3.1.3[65]. Associations between inverse-normalized protein levels and imputed dosages were tested using linear additive genetic regression models adjusted for age, sex, and body mass index[11].

For INTERVAL, linear regression models were performed on the follow-up SNPs using SNPTEST v2.5.2. Associations between inverse-normalized protein levels and imputed dosages were tested using linear additive genetic regression models adjusted for age, sex, duration between blood draw and sample processing, and the first three principal components of ancestry. As protein measurements were performed in two separate batches, associations were tested within each batch, and the results were meta-analyzed across batches using a fixed-effect inverse variance weighted meta-analysis in METAL.

**eQTL mapping**. We used linear mixed-effects models, accounting for familial relationships using "PEDIGREEMM" in R[65], to assess associations between ~8.5 million additively coded 1000 G SNPs and expression levels of 17,873 transcripts[19]. Models were adjusted for age, sex, platelet count, differential white cell count (percentages of lymphocyte, monocyte, eosinophil, and basophil), and 20 PEER factors[66,67] to reduce confounding from unmeasured factors. The criteria used to define *cis* and *trans* effects for pQTL variants were also applied to eQTL variants. A false discovery rate (FDR) threshold of 0.05 was applied separately for *cis*-eQTL and *trans*-eQTL variants.

**Colocalization analysis**. Colocalization analysis involved a two-step procedure. Using our *cis*-pQTL results, we first identified the locus that harbored the sentinel *cis*-pQTL variant for each protein. The locus was defined as 1 Mb region (upstream and downstream) of each sentinel *cis*-pQTL variant. Using FHS eQTL results[19], we then identified all genes within each locus for which expression of a gene was associated with the lead pQTL variant. Because eQTL analysis was based on 1000 Genomes-imputed SNPs, we used the lead pQTL variant from 1000 Genomes imputation when the lead pQTL variant was from Exome Chip. Second, because genes at the same locus are often correlated and regulated by the same *cis* SNPs, we only retained the gene with the lowest SNP-gene association $P$ value for each qualifying sentinel pQTL variant, which resulted in gene-protein pairs showing association with a common SNP and able to be tested for colocalization. To estimate the probability that *cis*-eQTLs and *cis*-pQTLs residing in the same genomic location shared the same causal variant, we conducted a Bayesian test for colocalization of all eQTL-pQTL pairs using the coloc package in R[20]. This method requires specifying a prior probability for a SNP being associated with gene expression only (p1), protein level only (p2), and with both traits (p12). We applied the default $P$ values, with p1 and p2 set to 1E−4, assuming that 1 in 10,000 SNPs are causal for either trait and p12 was set to 1E−5. We also used p1 and p2 values based on the number of eQTL and pQTL variants observed in our data. For the eQTL analysis, we detected 19,613 non-redundant eSNPs among 4,285,456 total *cis*-SNPs, indicating that the probability a SNP is a causal eSNP is 4.6E−3. This probability corresponds to the sum p1 + p12. For the pQTL analysis, we detected 254 non-redundant *cis*-pQTL variants among 440,409 total *cis*-SNPs, indicating that the probability that a SNP is a causal pQTL variant is 5.7E−4. This probability corresponds to the sum p2 + p12. p12 was set to 0.003 corresponding to a probability of 75% that a causal eSNP is also a causal pQTL variant, an approach that has been shown to represent the best choice[68].

**Mendelian randomization**. Leveraging the *cis*-pQTL variants identified in the current study, we used a two-sample MR approach to test for putatively causal associations between plasma proteins and CHD risk. Summary statistics for pQTL-CHD associations were from large meta-analyses of CARDIOGRAMplusC4D[1,69]. Pruned *cis*-pQTL variants (LD $r^2 < 0.1$) for each protein were used as instrumental variables (IVs) for the corresponding protein. For proteins with only one independent SNP after LD pruning, causal effect estimates were determined using the Wald ratio test, i.e., a ratio of effect per risk allele on CHD to effect per risk allele on inverse-rank normalized protein levels. When multiple non-redundant pQTL variants were present, we conducted multi-SNP MR using inverse-variance weighted estimates, i.e., a meta-result when using non-redundant pQTLs as an IV. All MR analyses were conducted using MRbase[70]. Causal effect estimates of proteins on CHD were interpreted per standard error increments in inverse-rank normalized protein level.

**Associations of protein levels with CVD**. To analyze associations between plasma protein levels and MI/CHD death and CVD death in FHS participants, protein biomarkers were rank-normalized. Cox proportional hazard models were used to predict MI/CHD death and CVD death for each biomarker after adjusting for age and sex. Participants younger than 50 years of age at baseline were excluded from outcome analyses due to a paucity of events in this age group. Participants with prevalent MI/CHD or CVD at baseline were also excluded from the longitudinal analyses, leaving a final sample size of 3520 FHS participants.

**Independent external replication**. After merging our 1000 G and Exome Chip GWAS results, the pQTL variant with the lowest $P$ value of association at each genetic locus was selected as the sentinel pQTL variant. We conducted independent external replication of our sentinel pQTL variants using the INTERVAL[52] and KORA studies[11] and prior protein GWAS. Of the 60 proteins with pQTLs in the FHS, replication was conducted for 48 proteins that were also measured in INTERVAL or KORA. The sentinel pQTL variant at each genetic locus in the FHS was determined to be successfully validated if its corresponding 1000G-imputed genotype or strong proxy (LD $r^2 > 0.8$) was also a significant pQTL variant for the corresponding protein in INTERVAL or KORA, and if directionality of pQTL-protein association was preserved. Statistical significance was defined as a $P$ value < $0.05/n$ ($n$ was the number of pQTL variants that were measured in replication cohorts).

**Study approval**. All participants from the FHS, INTERVAL, and KORA gave informed consent for participation in this study and for the collection of plasma and DNA for analysis. The FHS study protocol was approved by Boston Medical Center. The INTERNAL study protocol was approved by the National Research Ethics Service, UK (11/EE/0538). The KORA study protocol was approved by the Ethics Committee of the Bavarian Medical Association, Germany.

**Data availability**. The SNP, gene expression, protein expression data that support the findings from the FHS of this study have been deposited in dbGaP (dbGaP Study Accession: phs000363.v16.p10). The searchable pQTL results are accessible through the NCBI Molecular QTL Browser (https://preview.ncbi.nlm.nih.gov/gap/eqtl/studies/). Data for INTERVAL and KORA are available upon request and are subject to approval by the study review board.

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

## Acknowledgements

The Framingham Heart Study is funded by National Institutes of Health contract N01-HC-25195. This project was funded in part by the Division of Intramural Research, National Heart, Lung, and Blood Institute (NHLBI), National Institutes of Health (NIH), Bethesda, MD. The views expressed in this manuscript are those of the authors and do not necessarily represent the views of the National Heart, Lung, and Blood Institute; the National Institutes of Health; or the U.S. Department of Health and Human Services. JEH is supported in part by NIH grant K23-HL116780 and a Massachusetts General Hospital Hassenfeld Research Scholar Award. The KORA study was initiated and financed by the Helmholtz Zentrum München – German Research Center for Environmental Health, which is funded by the German Federal Ministry of Education and Research (BMBF) and by the State of Bavaria. Furthermore, KORA research was supported within the Munich Center of Health Sciences (MC-Health), Ludwig-Maximilians-Universität, as part of LMUinnovativ. The KORA Study Group consists of A. Peters (speaker), J. Heinrich, R. Holle, R. Leidl, C. Meisinger, K. Strauch, and their co-workers, who are responsible for the design and conduct of the KORA studies. The INTERVAL study is funded by NHSBT (11-01-GEN) and has been supported by the NIHR-BTRU in Donor Health and Genomics (NIHR BTRU-2014-10024) at the University of Cambridge in partnership with NHSBT. This study was partially funded by Merck and Co., Kenilworth, NJ, USA. The views expressed are those of the authors and not necessarily those of the NHS, the NIHR, the Department of Health of England, or NHSBT. The MRC/BHF Cardiovascular Epidemiology Unit is supported by the UK Medical Research Council (G0800270), British Heart Foundation (SP/09/002), UK National Institute for Health Research Cambridge Biomedical Research Centre, European Research Council (268834), and European Commission Framework Programme 7 (HEALTH-F2-2012-279233). B.B.S. is funded by the Cambridge School of Clinical Medicine MRC/Sackler

Prize PhD Studentship (MR/K50127X/1) and supported by the Cambridge School of Clinical Medicine MB-PhD programme. J.D. is a British Heart Foundation Professor, European Research Council Senior Investigator, and National Institute for Health Research (NIHR) Senior Investigator. KS was supported by 'Biomedical Research Program' funds at Weill Cornell Medicine in Qatar, a program funded by the Qatar Foundation.

## Author contributions

D.L., P.C. supervised the experiments. D.L., C.Y. conceived and designed the study. G.C., C.Y., J.K. wrote the manuscript. C.S., C.Y., S.-j.H., C.L., A.L. analyzed the data. B.B.S., A. L., J.C.M., C.G., J.G., J.D., H.R., A.S.B., K.S. conducted the replication analysis; Hs.W., J.E. H., P.C., M.G.L., A.D.J. contributed statistic advice, or analysis tools. All authors discussed the results and reviewed the final manuscript.

## Additional information

**Competing interests:** A.S.B. reports grants from Merck, Pfizer, Novartis, Biogen and Bioverativ and consulting fees from Novartis. J.D. sits on the Novartis Cardiovascular and Metabolic Advisory Board, and had grant support from Novartis. J.C.M. and H.R. were Merck employees at the time of their contributions to this study. The remaining authors declare no competing interests.

Chen Yao [1,2], George Chen[1,2], Ci Song[1,2,3,4], Joshua Keefe[1,2], Michael Mendelson[1,2,5], Tianxiao Huan[1,2], Benjamin B. Sun [6], Annika Laser[7,8], Joseph C. Maranville[9], Hongsheng Wu[10], Jennifer E. Ho[11], Paul Courchesne[1,2], Asya Lyass[1,12], Martin G. Larson[1,13], Christian Gieger[7,8,14], Johannes Graumann [15], Andrew D. Johnson[1,2], John Danesh[6,16,17], Heiko Runz[9], Shih-Jen Hwang[1,2], Chunyu Liu[1,2], Adam S. Butterworth [6,18], Karsten Suhre [19] & Daniel Levy[1,2]

[1]Framingham Heart Study, Framingham 01702 MA, USA. [2]Population Sciences Branch, Division of Intramural Research, National Heart, Lung, and Blood Institute, National Institutes of Health, Bethesda 20892 MD, USA. [3]Department of Medical Sciences, Uppsala University, 75105 Uppsala, Sweden. [4]Department of Immunology, Genetics and Pathology, Uppsala University, 75105 Uppsala, Sweden. [5]Department of Cardiology, Boston Children's Hospital, Boston 02115 MA, USA. [6]MRC/BHF Cardiovascular Epidemiology Unit, Department of Public Health and Primary Care, University of Cambridge, Cambridge CB1 8RN, UK. [7]Research Unit of Molecular Epidemiology, Helmholtz Zentrum München, German Research Center for Environmental Health, Ingolstädter Landstraße 1, 85764 Neuherberg, Germany. [8]Institute of Epidemiology II, Helmholtz Zentrum München, German Research Center for Environmental Health, Ingolstädter Landstraße 1, 85764 Neuherberg, Germany. [9]MRL, Merck & Co., Inc, Kenilworth 07033 NJ, USA. [10]Computer Science and Networking, Wentworth Institute of Technology, Boston 02115 MA, USA. [11]Cardiovascular Research Center and Division of Cardiology, Department of Medicine, Massachusetts General Hospital, Boston 02114 MA, USA. [12]Department of Mathematics and Statistics, Boston University, Boston 02115 MA, USA. [13]Department of Biostatistics, Boston University School of Public Health, Boston 02118 MA, USA. [14]German Center for Diabetes Research (DZD), Ingolstädter Landstraße 1, 85764 Neuherberg, Germany. [15]Scientific Service Group Biomolecular Mass Spectrometry, Max Planck Institute for Heart and Lung Research, W.G. Kerckhoff Institute, Ludwigstr. 43, D-61231 Bad Nauheim, Germany. [16]British Heart Foundation Cambridge Centre of Excellence, Division of Cardiovascular Medicine, Addenbrooke's Hospital, Cambridge CB2 0QQ, UK. [17]Department of Human Genetics, Wellcome Trust Sanger Institute, Wellcome Trust Genome Campus, Hinxton, Cambridge CB10 1RQ, UK. [18]NIHR Blood and Transplant Research Unit in Donor Health and Genomics, Department of Public Health and Primary Care, University of Cambridge, Cambridge CB1 8RN, UK. [19]Department of Physiology and Biophysics, Weill Cornell Medicine-Qatar, Education City, PO 24144 Doha, Qatar. These authors contributed equally: Chen Yao, George Chen, Ci Song.

