## [Peer Review File · Nature Communications]

Reviewer #1 (Remarks to the Author):

Yao et al present a revised manuscript to address concerns in the first round of review.

A very helpful addition is Figure 1. While my own taste would have some of those numbers changed (e.g., rather than total number of associations which most don't care about, the number of physical loci tagged by LD-independent sentinel SNPs), this is helpful to focus on what's potentially the reported outputs from the paper.

The challenge I have on this read is that each of the presented analyses don't feel "tight" or "conclusive" (save the initial discovery, minus the relatively large number of trans-pQTLs) around any given story. Each individual unit has some challenges that aren't closed up:

- "enrichment with eQTLs" — this represents enrichment, not formal co-localization. Chance overlap is very likely to happen, so the numbers there aren't surprising nor do they tell us about specific pathways or biology

- "functional, reg, clinical enrichments - enrichment of CAD pathways for a preselected set of proteins screened for CHD relevance is circular. The chance overlap with coding variants isn't explored deeply - the work on page 5 is really high level or confirmatory reporting.

- "MR" - done with trans-QTLs which is a difficult set to explore. The genes highlighted and connection to epidemiological correlation is not crisp. The genes that are implicated here don't have an obvious, biological story.

- "pQTL overlap with CHD SNP" - again, we care about co-localized results, not chance overlaps. You'd expect some of this. No "tight" inference directed to a biological inference.

The above issues are placed in a context of a /large/ number of trans-pQTL discoveries (more numerous than cis-pQTLs), which even after a more stringent testing correction still makes me concerned.

This is not to say that there aren't potential good things in the paper - the discovery and efforts to replicate are laudable; multiple studies with different platforms for assessment of variation. It is definitely believable that there exists pQTLs in there.

Comments.

1. power calcs. Given the size of the replication study, and the given effect sizes (which are likely winner's cursed, which is central to the authors' contention), it should be knowable the extent to which replication was expected. Does the given number match these expectations, exceed, or are less than?

The author's response here — that lower replication is due to loss of power from either (i) effect sizes, or (ii) sample size are essentially the 'same coin' by and large. But that is knowable, to some extent, given the replication attempt they did employ. Does the % yield make sense?

If predicted power was high for all attempts, then it begs the question if the criteria for significance is not stringent enough, or that there's something else that is not able to be understood.

2. eQTL/pQTL overlap.

2a. Because it now seems clear that most variants (at least, associated with GWAS are eQTLs), are eQTLs, the 'enrichment' of eQTLs around pQTLs is a much less interesting observation.

A more interesting, useful question - to what extent do pQTLs /co-localize/ their statistical association with pQTLs?

In the previous review, I asked the authors to perform co-localization experiments. They state that 5,239 (~40% of cis-pQTLs) 'co-localized' with an eQTL. I'm looking at that table, and it's not clear where, if any, formal "co-localization" analysis has been performed. See existing tools: COLOC, eCaviar, etc. Perhaps this is most interesting to do with genome-wide significant pQTLs.

2b. In addition, here you are also potentially implicating a specific gene. So the question is not just of statistical co-localization, but also that the same gene is implicated. At face, the fact that only

11/114 of the sentinel pQTL variants ping the same gene probably begs some further scrutiny. If the tissues are approximately the same assayed, shouldn't this be higher? What other ways in which a pQTL could be biologically bona-fide, yet not generate an eQTL signature in the appropriate tissue?

2c. for the 11/14, there seems like a majority are directionally consistency (I read: 8/11), but shouldn't this be substantially higher/near perfect? That is, unless the pQTLs and eQTLs aren't co-localized. Let's say all of these 11 co-localized. Can the authors describe a model for which this inconsistency of directionality makes sense?

3. CHD overlaps (comments 6a).

3a. The authors state that the "localization" was performed "by direct matching" which isn't exactly what one wants here. As above, how many of the pQTLs actually co-localize with CHD associations? COLOC and other statistical tools here will tell you if the distribution of associations signals for both scans "line up", implicating the same causal variant. The implications is that two associations partially overlap, but don't implicate the same causal variant and/or gene, which could be coincidental.

4. trans-QTL effects. The number of effects here — measured as a function of the 'sentinel' loci - is 2:1 that of the cis-eQTLs. this doesn't feel right. Empirically, at least in gene expression space, we find far more cis-eQTL effects than trans-eQTLs. I have to believe that a similar intuition applies here. Put another way: not all proteins measured have a detectable cis-pQTLs (at most 50%, 32 of / 71), which to me is surprising.

5. The Causal inference / MR (+ epidemiological correlation) experiments, linking pQTLs to gene loci

5a. These results (based on trans-QTLs which are undoubtedly amongst the most likely to have additional, pleiotropic effects) are perhaps the least optimal to include in an MR experiment for an outcome. intrinsically, these are the hardest to believe on face.

5b. An example as stated earlier is the CELSR2, APOB relationship. surely this must reflect

5b. But then Figure 5 attempts to put together the MR effect sizes with the associated hazard in FHS. The story to take back from this plot is challenging. You have a ton of things that are null (mostly the

Epi HRs) but then opposite direction (PON1, BCHE, sRAGE). The positive controls are certainly working (LPA), and maybe there a signal there - but it looks very noisy.

minor comments

- how were the statistical threshold determined? $1.2E-7$ for cis pQTLs? the $7E-10$ I savvy, given the back-of-envelope I suggested to the authors. But in the methods, it should be clear exactly how they arrived at the threshold they did.

Reviewer #2 (Remarks to the Author):

Yao et al. have much clarified and improved the manuscript, particularly emphasising the novel results, and those that had replication. It is now suitable for publication, in my opinion. The integration of the pQTL data with eQTLs in the NCBI database will be very useful for researchers to browse and to take the novel findings forward, and in my opinion is the main strength of the study. I look forward to seeing the published version.

Detailed Response

Reviewer #1 (Reviewer's comments are presented in black, Authors' replies are in blue):

Yao et al present a revised manuscript to address concerns in the first round of review.

A very helpful addition is Figure 1. While my own taste would have some of those numbers changed (e.g., rather than total number of associations which most don't care about, the number of physical loci tagged by LD-independent sentinel SNPs), this is helpful to focus on what's potentially the reported outputs from the paper.

The challenge I have on this read is that each of the presented analyses don't feel "tight" or "conclusive" (save the initial discovery, minus the relatively large number of trans-pQTLs) around any given story. Each individual unit has some challenges that aren't closed up:

- "enrichment with eQTLs" — this represents enrichment, not formal co-localization. Chance overlap is very likely to happen, so the numbers there aren't surprising nor do they tell us about specific pathways or biology
- "functional, reg, clinical enrichments - enrichment of CAD pathways for a preselected set of proteins screened for CHD relevance is circular. The chance overlap with coding variants isn't explored deeply - the work on page 5 is really high level or confirmatory reporting.
- "MR" - done with trans-QTLs which is a difficult set to explore. The genes highlighted and connection to epidemiological correlation is not crisp. The genes that are implicated here don't have an obvious, biological story.
- "pQTL overlap with CHD SNP" - again, we care about co-localized results, not chance overlaps. You'd expect some of this. No "tight" inference directed to a biological inference.

The above issues are placed in a context of a /large/ number of trans-pQTL discoveries (more numerous than cis-pQTLs), which even after a more stringent testing correction still makes me concerned.

This is not to say that there aren't potential good things in the paper - the discovery and efforts to replicate are laudable; multiple studies with different platforms for assessment of variation. It is definitely believable that there exists pQTLs in there.

Reply:

We thank the reviewer for the complimentary comments and useful suggestions for further improvement. As detailed below, we have made numerous revisions in response to the reviewer's suggestions:

1. Based on reviewers' comments, we have raised the SNP imputation quality r^2 from 0.3 to 0.5. Our number of pQTL variants decreased from 17,893 SNPs associated with 60 proteins to 16,602 SNPs for 57 proteins. Three proteins (Adipsin, IGFBP3, SDF1) were removed, because of low pQTL variant

imputation quality. The number of sentinel pQTL variants (representing non-redundant genetic loci) decreased from 114 to 103. We have revised multiple sections of the manuscript accordingly.

“pQTL Mapping: Using Bonferroni correction for multiple testing, we identified 16,602 pQTL variants (with Reference SNP cluster IDs) associated with 57 proteins (Table S3), including 11,806 cis-pQTL variants (at $P < 1.25E-7$) for 40 proteins and 4,796 trans-pQTL variants (at $P < 7.04E-10$) for 44 proteins; 27 proteins had both cis- and trans-pQTL variants. Our study had 80% power to detect a cis- or trans-pQTL variant that explained $\geq 0.6\%$ of variance in protein levels (Table S4). Pruning of the pQTL variants (linkage disequilibrium [LD] $r^2 < 0.1$) yielded 372 non-redundant variants (Table S5) representing 103 sentinel loci (the variant with the lowest pQTL-protein association P value, Table S6) consisting of 40 sentinel cis-pQTLs (Figure 2a) and 63 sentinel trans-pQTLs (Figure 2b). Among the 16,602 pQTL variants, 341 were coding variants associated with 19 proteins (Table S7) and 33 were rare variants (minor allele frequency $< 1\%$ genotyped on Exome Chip) associated with 17 proteins (Table S8). In addition, 1,689 insertion/deletion polymorphisms were identified for 55 proteins (Table S9).”

“Integrating pQTL Variants with CHD-associated SNPs: We integrated our pQTL variants with 2,213 CHD-related SNPs from the CARDIoGRAMplusC4D Consortium GWAS¹⁻³. A total of 210 pQTL variants (16 non-redundant variants at LD $r^2 < 0.1$ representing 9 proteins; Table S13) exactly matched SNPs associated with CHD from prior GWAS. Table 1 displays the sentinel pQTL variants that coincided with CHD-related GWAS SNPs and the corresponding protein at each genetic locus. The proteins with pQTL variants coinciding with CHD-associated SNPs included LPA, APOB, B2M, CRP, GMP140, GRN, MCAM, sGP130, and sICAM1. It is important to note, however, that these results do not indicate a causal relationship between the pQTL-associated protein and CHD.”

“Trans-pQTL variants and CHD: Of the nine proteins with pQTL variants that coincided with CHD-associated SNPs, eight had pQTLs with trans effects (54 non-redundant trans-pQTL variants in total) and 69% of these trans-pQTL variants were also associated with the expression of nearby genes (cis-eGenes), i.e. these trans-pQTL variants were also cis-eQTL variants associated with the expression of nearby cis-eGenes. Based on these findings, we hypothesized that trans-pQTL variants may regulate circulating protein levels through cis-effects on the expression of nearby cis-eGenes (Figure S3). To test this hypothesis, we employed Mendelian randomization (MR)¹⁶ using the expression of all genes within 1 Mb from the trans-pQTL locus as the exposure, cis-eQTLs associated with these genes (from the FHS whole blood gene expression database²⁰) as instrumental variables, and circulating protein levels as the outcome. We found that the effects of trans-pQTLs on circulating protein levels were causally regulated by expression of cis-eGenes for all eight proteins (Table S14). To extend these findings to other CHD-related tissues, we applied the same analyses to GTEx²⁵ whole blood, liver, and heart eQTLs. For two of the proteins (APOB and GRN), there was additional experimental evidence in support of our results through interrogation of GTEx²⁶ whole blood eQTLs (Table S14). Moreover, we found significant causal effects of PSRC1 expression on APOB levels (in liver), PSRC1 on GRN levels (in artery), and ABO on GMP140 levels (in heart atrial appendage).”

“Causal Testing: We applied MR testing using pruned cis-pQTL variants ($LD r^2 < 0.1$) as instrumental variable for circulating protein levels in order to identify proteins that were causal for CHD. MR testing was conducted for all 40 proteins with cis-pQTLs and causally implicated LPA, BCHE, PON1, MCAM, MPO, and cystatin C ($P < 0.05$; Table S15). Causal testing for LPA and BCHE remained statistically significant after adjusting for multiple testing ($P < 0.05/40$). “

“Protein Associations with Clinical Outcomes: For the 13 proteins with pQTL variants that either coincided with CHD GWAS SNPs (9 proteins) or tested positive by MR (6 proteins) at $P < 0.05$, we sought to determine the longitudinal associations of circulating levels of these proteins with a) major CHD events (recognized myocardial infarction or CHD death; $n = 213$ events) and b) CVD death (fatal CHD or death due to stroke, peripheral arterial disease, heart failure, or other CVD causes; $n = 199$ events) with a median follow-up of 14.3 years (25th percentile 11.4, 75th percentile 15.2 years) among 3,520 FHS participants 50 years of age or older. Ten of the 12 proteins were nominally associated ($P < 0.05$) with incident CHD and/or CVD death (Table 2), and eight proteins remained statistically significant after adjusting for multiple testing ($P < 0.05/13$). Two of the six proteins (PON1 and cystatin C) that tested causal for CHD by MR at $P < 0.05$ were also associated with long-term CHD/CVD outcomes at $P < 0.0038$. The protein effect sizes on CHD predicted from MR were directionally consistent with the observed prospective protein-CHD associations in all cases except for PON1 (Figure 5) “

2. Figure 1: We have clarified our flow diagram by stating the total number of physical loci identified: “GWAS of 71 proteins yielded 16,602 pQTL variants representing 103 loci for 57 proteins.” Please see revised Figure 1 below:

3. Enrichment with eQTLs: We have replaced the “Enrichment of pQTLs with eQTLs” section with “Colocalization of pQTLs and eQTLs”. Please refer to our new analyses and our reply to Comment 2.
4. Functional, regulatory, and clinical enrichments: We conducted pathway enrichment for each protein using the annotated genes corresponding to its pQTL variants. We did not conduct pathways enrichment using the preselected proteins because they represent a biased set of proteins with a high prior probability of association with cardiovascular disease.
5. MR: We did not use *trans*-QTL variants as instrumental variables in any MR analyses. We only used *cis*-eQTL variants or *cis*-pQTL variants. Please refer to our detailed reply to Comment 5.
6. pQTL overlap with CHD SNPs: We have conducted a new colocalization analysis of pQTLs and CHD GWAS SNPs. Please refer to our reply to Comment 3.

Reviewer 1 Comments.

Comment 1. Power calcs. Given the size of the replication study, and the given effect sizes (which are likely winner’s cursed, which is central to the authors’ contention), it should be knowable the extent to which replication was expected. Does the given number match these expectations, exceed, or are less than? The author’s response here — that lower replication is due to loss of power from either (i) effect sizes, or (ii) sample size are essentially the ‘same coin’ by and large. But that is knowable, to some extent, given the replication attempt they did employ. Does the % yield make sense? If predicted power was high for all attempts, then it begs the question if the criteria for significance is not stringent enough, or that there's something else that is not able to be understood.

Reply: We thank the reviewer for this suggestion. We have performed 1000 re-samplings of 3300 FHS participants to evaluate our power for replication. We predicted that there would be 80 sentinel pQTL-protein associations with 80% power to replicate. We found that 54 (67%) of them replicated in the external INTERVAL cohort. The failure of proteins to replicate may be due proteomic platform differences between discovery and replication studies.

We have modified the Methods section as follows:

“Power Calculation: For an estimate of empirical power in the replication stage (sample size of approximately 3300 in INTERVAL), we performed pQTL analysis with 1,000 resamplings of 3,300 FHS participants. Half of them were unrelated and the rest were randomly sampled. We counted the number of results with $P < 0.05/n$ in the 1,000 resamplings, where n is the number of pQTL variants that were tested for replication in INTERVAL.”

We have modified the Results section as follow:

*“External Replication: Among our 103 sentinel pQTLs linked to 57 proteins, 96 sentinel pQTLs (36 *cis*- and 60 *trans*-pQTLs) associated with 51 proteins were not previously reported in GWAS. Therefore, we attempted to replicate all 103 sentinel pQTLs in the INTERVAL¹⁵ (N=3301) and the KORA¹¹ studies (N=997). Among our 57 proteins linked to 103 sentinel pQTLs, 45 proteins (associated with 32 sentinel *cis*-pQTLs and 56 sentinel *trans*-pQTLs) were independently measured*

in the INTERVAL study. Of the 32 sentinel cis-pQTL-protein pairs (for 32 proteins) from the FHS, 21 (66%) replicated in INTERVAL at $P < 5.7E-4$ (alpha level of 0.05 after Bonferroni correction for 88 tests; $0.05/88$). Of the 56 sentinel trans-pQTL-protein pairs (for 37 proteins) from the FHS, 33 (59%) trans-pQTL-protein pairs (for 22 proteins) replicated in INTERVAL ($P < 5.7E-4$; Table S10). One additional trans-pQTL-protein pair replicated in KORA. Four proteins (associated with four cis-pQTLs and one trans-pQTL) that were not measured or did not replicate in INTERVAL or KORA replicated based on prior GWAS evidence (Table S10). The remaining 10 proteins without any available external source of replication were associated with six cis- and six trans-pQTLs (Table S10). In total, 24 (71%) sentinel cis-pQTLs and 35 (61%) sentinel trans-pQTLs replicated (at Bonferroni corrected $P < 0.05$) with 100% consistent direction of effect compared with the FHS discovery results.”

“Resampling analysis: Based on 1,000 re-samplings of 3,300 FHS participants in the discovery sample, 80 pQTL-protein associations (31 cis and 49 trans) yielded $P < 5.7E-4$ (alpha level of 0.05 after Bonferroni correction for multiple testing; $0.05/88$) in $\geq 80\%$ of samplings and thus were considered likely to replicate in a GWAS sample size of 3,300 individuals from INTERVAL (Table S10). Among the 80 pQTL-protein associations that were considered likely to replicate, 54 (68%) replicated in INTERVAL. The discrepancy between predicted and observed replication may be due in part to proteomic platform differences between the discovery and replication studies.”

Comment 2. eQTL/pQTL overlap.

2a. Because it now seems clear that most variants (at least, associated with GWAS are eQTLs), are eQTLs, the ‘enrichment’ of eQTLs around pQTLs is a much less interesting observation. A more interesting, useful question - to what extent do pQTLs /co-localize/ their statistical association with pQTLs? In the previous review, I asked the authors to perform co-localization experiments. They state that 5,239 (~40% of cis-pQTLs) ‘co-localized’ with an eQTL. I’m looking at that table, and it’s not clear where, if any, formal “co-localization” analysis has been performed. See existing tools: COLOC, eCaviar, etc. Perhaps this is most interesting to do with genome-wide significant pQTLs.

Reply: We thank the reviewer for this suggestion. We have conducted de novo colocalization analyses using the COLOC package as the reviewer suggested. Among the 33 proteins that have both cis-pQTLs and cis-eQTLs, 19 proteins were found to have >75% probability of sharing a causal SNP.

We have modified the Statistical Methods section as follows:

“Colocalization analysis: For the colocalization analyses, we first pruned our pQTL variants, retaining only those with LD $r^2 < 0.1$. Using FHS eQTL results²⁰, we then identified transcripts that were also associated with the same pQTL variant (i.e. pQTL variant = eQTL variant). To estimate the probability that cis-eQTLs and cis-pQTLs residing in the same genomic location shared the same causal variant, we conducted a Bayesian test for colocalization of all eQTL-pQTL pairs using the coloc package in R²¹. This method requires specifying a prior probability for a SNP being associated with gene expression only (p1), protein level only (p2), and with both traits (p12). We

applied the default P values, with p1 and p2 set to 1E-4, assuming that 1 in 10,000 SNPs are causal for either trait and p12 was set to 1E-5. We also used p1 and p2 values based on the number of eQTL and pQTL variants observed in our data. For the eQTL analysis, we detected 19,613 non-redundant eSNPs among 4,285,456 total *cis*-SNPs, indicating that the probability a SNP is a causal eSNP is 4.6E-3. This probability corresponds to the sum p1 + p2. For the pQTL analysis, we detected 254 non-redundant pQTL variants among 440,409 total *cis*-SNPs, indicating that the probability that a SNP is a causal pQTL variant is 5.7E-4. This probability corresponds to the sum p1 + p2. p12 was set to 0.003 corresponding to a probability of 75% that a causal eSNP is also a causal pQTL variant, an approach that has been shown to represent the best choice⁷⁷."

We modified the Results section as follows:

"Colocalization of pQTLs and eQTLs: Among the 372 non-redundant pQTL variants, we identified 190 unique variants (associated with 53 proteins) that were also eQTL variants (genetic variants associated with whole blood gene expression levels in FHS participants²⁰) at FDR<0.05. These 190 eQTL variants consisted of 188 cis-eQTL variants and 27 trans-eQTL variants (Table S11), suggesting that a substantial number of causal eQTL variants may also be causal pQTLs. To test this hypothesis, we conducted a Bayesian test of colocalization of pruned cis-pQTL variants (LD r^2 <0.1) using the coloc package in R²¹ (see Methods). A total of 117 of 254 unique cis-pQTL variants (associated with 33 proteins) were tested for colocalization with gene expression for 136 transcripts residing within 1 Mb of the cis-pQTL variant (FDR <0.05). Using transcripts and proteins associated with a shared SNP, we conducted a colocalization test for each protein to determine the probability that the two association signals were due to the same causal variant. The prior probabilities for a SNP being associated with gene expression only (p1), protein level only (p2), or with both traits (p12) were based on the number of eSNPs and pQTL variants observed in our data (see Methods). The value for p12 was set to 75%, i.e. the probability that a causal eSNP is a causal pQTL variant. For 19 out of 33 proteins that were associated with both cis-pQTL variants and eQTL variants, we observed a probability >75% that the pQTL variants colocalized with the eQTL variants (Table S12). We generated colocalization results for 19 proteins by applying default P values (p1=1E-4, p2=1E-4, p12=1E-5) in the coloc package assuming that 1 in 10,000 SNPs is causal for either trait."

Table S12. Proteins with >75% probability of co-localization of pQTL variants with eQTLs variants (PP.H4 value >0.75 represents co-localization of pQTL and eQTL causal variants)

Protein	#SNPs	PP.H0	PP.H1	PP.H2	PP.H3	PP.H4
GMP140	7	2.32E-196	7.06E-85	3.50E-115	0	1
ADM	1	1.41E-54	7.14E-14	2.10E-44	0	1
KLKB1	2	1.63E-130	1.30E-20	1.34E-113	1.91E-13	1
sGP130	1	2.12E-29	1.07E-12	2.10E-20	0	1
COL18A1	1	5.32E-80	8.33E-12	6.81E-72	0	1
FGG	1	2.83E-16	3.03E-10	9.99E-10	0	1
AGP1	2	2.63E-20	1.02E-11	2.75E-12	1.19E-08	1

Ceruloplasmin	1	4.61E-19	1.20E-14	4.10E-08	0	1
A1M	1	9.25E-17	5.44E-08	1.81E-12	0	1
EFEMP1	3	1.69E-35	2.52E-12	7.16E-27	6.54E-08	1
GP5	1	7.41E-18	8.20E-14	9.64E-08	0	1
GRN	1	2.51E-17	4.38E-15	6.13E-06	0	0.999994
CD40L	1	2.79E-09	2.32E-07	1.28E-05	0	0.999987
NCAM	1	1.85E-11	1.22E-09	1.62E-05	0	0.999984
MMP9	1	6.67E-19	2.29E-17	3.11E-05	0	0.999969
MCAM	1	3.89E-07	3.38E-06	0.000122677	0	0.999874
MMP8	1	2.77E-17	3.64E-17	0.000812252	0	0.999188
CXCL16	2	9.00E-15	2.74E-10	3.92E-06	0.11847	0.881526
UCMGP	4	6.37E-45	5.61E-39	2.66E-07	0.233573	0.766426

PP.H0 (no causal variant), PP.H1 (causal variant for gene expression only), PP.H2 (causal variant for protein only), PP.H3 (two distinct causal variants), PP.H4 (one common causal variant)

2b. In addition, here you are also potentially implicating a specific gene. So the question is not just of statistical co-localization, but also that the same gene is implicated. At face, the fact that only 11/114 of the sentinel pQTL variants ping the same gene probably begs some further scrutiny. If the tissues are approximately the same assayed, shouldn't this be higher? What other ways in which a pQTL could be biologically bona-fide, yet not generate an eQTL signature in the appropriate tissue?

Reply: Although pQTLs and eQTLs were both identified in FHS participants, gene expression and protein expression were measured at different time points, on average 6-7 years apart, as described in the Methods section. To avoid misinterpretation, we have replaced the "Enrichment of pQTLs with eQTLs" section with "Colocalization of pQTL variants and eQTL variants". Please see the reply to Comment 2a for details.

2c. For the 11/14, there seems like a majority are directionally consistency (I read: 8/11), but shouldn't this be substantially higher/near perfect? That is, unless the pQTLs and eQTLs aren't co-localized. Let's say all of these 11 co-localized. Can the authors describe a model for which this inconsistency of directionality makes sense?

Reply: Because gene expression and protein expression were measured at different time point, we are reluctant to compare directional consistency between gene expression and plasma protein levels. As the reviewer suggested, we have replaced the "Enrichment of pQTLs with eQTLs" section with "Colocalization of pQTL variants and eQTL variants". Please see our reply to Comment 2a for details.

Comment 3. CHD overlaps (comments 6a).

3a. The authors state that the "localization" was performed "by direct matching" which isn't exactly what one wants here. As above, how many of the pQTLs actually co-localize with CHD associations? COLOC and other statistical tools here will tell you if the distribution of associations signals for both scans "line up", implicating the same causal variant. The implications is that two associations partially overlap, but don't implicate the same causal variant and/or gene, which could be coincidental.

Reply: We thank the reviewer for this suggestion. Colocalization analysis can only be applied to *cis*-pQTL variants that overlap with GWAS SNPs for CHD (at $P < 5e-8$). The only protein eligible for colocalization analysis using *cis*-pQTL variants was LPA. The co-localization result for LPA was highly significant, as shown below. Note that for all other proteins with pQTL variants that coincided with GWAS SNPs for CHD, the pQTL variants were exclusively *trans*-pQTLs and colocalization could not be performed. Given the scarcity of proteins eligible for co-localization analysis, we decided to not present the results for LPA (for which we instead share the more informative MR results). The colocalization results for LPA are shown below:

Protein	#SNPs	PP.H0.abf	PP.H1.abf	PP.H2.abf	PP.H3.abf	PP.H4.abf
LPA	5	1.03E-182	2.95E-36	1.15E-150	7.25E-13	1

4. *trans*-QTL effects. The number of effects here — measured as a function of the ‘sentinel’ loci - is 2:1 that of the *cis*-eQTLs. this doesn’t feel right. Empirically, at least in gene expression space, we find far more *cis*-eQTL effects than *trans*-eQTLs. I have to believe that a similar intuition applies here. Put another way: not all proteins measured have a detectable *cis*-pQTLs (at most 50%, 32 of / 71), which to me is surprising.

Reply: The reviewer is comparing *cis*-eQTLs with *cis*-pQTLs, but SNP effects on gene expression may be quite different from those on protein expression. We found that *trans*-pQTL variants can regulate remote proteins through the expression of nearby *cis*-eGenes (Please refer to the “*Trans*-pQTLs and CHD” section in manuscript for details). Therefore, it may not be surprising that we observed fewer *cis* effects than *trans* effect. To address this further, we checked on the *cis* vs. *trans* associations reported in the INTERVERAL study (<https://www.biorxiv.org/content/early/2017/05/05/134551>), which assessed pQTL variants for ~3000 proteins. We discovered that they had very similar results to ours. They found 1478 proteins to have pQTLs. Among them, 374 (25%) proteins had *cis*-pQTLs only, 925 (63%) proteins had *trans*-pQTLs only, and 179 (12%) proteins had both *cis*- and *trans*-pQTLs. Therefore, we concluded that our *cis*-to-*trans* ratio, while different from that for eQTLs, is consistent with that from the INTERVAL study.

5. The Causal inference / MR (+ epidemiological correlation) experiments, linking pQTLs to gene loci

5a. These results (based on *trans*-QTLs which are undoubtedly amongst the most likely to have additional, pleiotropic effects) are perhaps the least optimal to include in an MR experiment for an outcome. intrinsically, these are the hardest to believe on face.

Reply: We agree with the reviewer that *trans*-QTLs are more likely to violate the assumptions of MR testing, due to potential pleiotropic effects. The same concern had been raised in the last round of review. Therefore, in the submitted manuscript, we updated all MR analyses using only *cis*-pQTLs as instrumental variables for each protein. In the current manuscript, we further modified the Methods and Results sections to emphasize this information more clearly.

We have modified the Statistical Methods section as follows:

“Mendelian randomization: Leveraging the cis-pQTL variants identified in the current study, we used a two-sample MR approach to test for putatively causal associations between plasma proteins and CHD risk. Summary statistics for pQTL-CHD associations were from large meta-analyses of CARDIOGRAMplusC4D^{1,78}. Pruned cis-pQTLs ($LD\ r^2 < 0.1$) for each protein were used as instrumental variables (IVs) for the corresponding protein. For proteins with only one independent SNP after LD pruning, causal effect estimates were determined using the Wald ratio test, i.e., a ratio of effect per risk allele on CHD to effect per risk allele on inverse-rank normalized protein levels. When multiple non-redundant pQTLs were present, we conducted multi-SNP MR using inverse-variance weighted estimates, i.e., a meta-result when using non-redundant pQTLs as an IV. All MR analyses were conducted using MRbase.⁷⁹ Causal effect estimates of proteins on CHD were interpreted per standard error increments in inverse-rank normalized protein level.”

5b. An example as stated earlier is the CELSR2, APOB relationship. surely this must reflect

Reply: As noted above in our reply to Comment 5a, we only used *cis*-pQTLs to explore the causal associations between protein levels and CHD. As we did not identify any *cis*-pQTL variants for APOB in the current study, we did not further explore the causal effect of circulating APOB levels on CHD. However, we considered it interesting to understand the underlying regulatory mechanism between *trans*-pQTLs and protein levels. Therefore, we used *cis*-eQTL variants for genes residing within the CELSR2/SORT1/PSRC1 locus as instrumental variables to test if circulating levels of APOB are causally affected by *cis*-eGene expression. We found that PSRC1 in the CELSR2/SORT1/PSRC1 locus causally affected APOB protein levels. Please refer to Figure S3 and the “*Trans*-pQTLs and CHD” section in the manuscript for details.

5b. But then Figure 5 attempts to put together the MR effect sizes with the associated hazard in FHS. The story to take back from this plot is challenging. You have a ton of things that are null (mostly the Epi HRs) but then opposite direction (PON1, BCHE, sRAGE). The positive controls are certainly working (LPA), and maybe there a signal there - but it looks very noisy.

Reply: We thank the reviewer for this comment. When interpreting the results from the MR test, we considered it worthwhile to compare the putative causal estimates with the observed protein-trait associations, taking advantage of the longitudinal cardiovascular disease outcome data available in the FHS. When there was an overlap between the confidence intervals of the MR odds ratio and the longitudinal protein-trait hazards ratio, we considered the putative causal estimate and the observed protein-trait estimate to be not significantly different. We therefore included both estimates in Figure 5 to allow such comparisons. When the 95% CI of the MR odds ratio and the longitudinal protein-trait hazards ratio did not overlap, the putative causal estimate and the observed protein-trait association estimate were considered to be significantly different. This was the case only for PON1, where the putative causal estimate and observed protein-trait estimate differed significantly. We further addressed a potential explanation for this unexpected difference in the “Novel Proteins and Pathways Implicated in CHD” section.

minor comments

- how were the statistical threshold determined? $1.2E-7$ for cis pQTLs? the $7E-10$ I savvy, given the back-of-envelope I suggested to the authors. But in the methods, it should be clear exactly how they arrived at the threshold they did.

Reply: We thank the reviewer for this suggestion. We have revised the Methods section as follows:

“We estimated that there were 440,409 potential *cis* SNP-protein pairs in total. Therefore, the Bonferroni-corrected P value for *cis*-pQTL variants was calculated as $0.05/440,409 = 1.25E-7$. The number of potential *trans* SNP-protein pairs was 8.5 million (SNPs) x 71 (proteins) - 440,409, yielding a Bonferroni corrected P value for *trans*-pQTLs of $7.04E-10$.”

Reviewer #1 (Remarks to the Author):

The authors provide a details response to my concerns. I have additional comments here.

1. Functional, reg, clinical enrichments.

a. In my previous comments, I suggested that because the enrichment analysis based on the proteins pre-selected to related to CHD, any enrichment analysis performed here is to be expected.

In looking more closely at this, it doesn't appear that there's any statistical quantification of enrichment at all here (w.r.t. gene enrichment). The use of the specific tool is simply as one for annotation. The annotations w.r.t. gene location are fine.

As such, I'm not sure at all what the the last 3 sentences add, or Figure 2. If the genes selected for protein quantification are already CVD related, then all of this is known, enrichments are not tested, so I don't know wha any of this actually adds.

b. It looks to me like the KEGG Pathway analysis involves taking all the genes in the interval around a pQTL — not sure this realistically adds anything at all.

c. It also seems like most of the supplement (Figure S2) is a long list of these outputs — again, I'm not sure 100+ pages of these figures really add much and certainly aren't particularly useful.

2. Co-localization analysis. The authors respond by performing a colocalization analysis with coloc. The authors state that the first thing they did was “pruned the pQTL variants, retaining only those with $LD\ r^2 < 0.1$ ”. I'm not entirely sure I understand what this means; the specifics for priors I understand (could be left in methods), but the overall approach as worded is unclear enough to me that I want to make sure this analysis is performed in the way that I expect it to.

Can the author confirm the following general approach:

a. Identify a locus that harbors a pQTL - e.g., at least one sentinel SNP that is the most strongly associated, surpassing a multiple-test correction. Note the targeted associated gene of interest.

b. Determine if, for the given target gene, if there exists an associated eQTL for that gene in FHS (blood).

c. If there exists at least one eQTL associated variant in the interval, perform co-localization, i.e.

d. Take the regional association data for all variants (say, that spans the territory), for pQTL and eQTL association. This step should involve all SNP associations data, not pruned data (looking at Table S12, not sure what n(SNP) means — this can't be the number of SNPs going into the coloc analysis?)

e. Report PP1, PP2, PP3, PP4 data. Ideally, $PP4/(PP3+PP4)$ suggests some localization, very high PP4 values denote very strong association

f. Visual (heuristic) conformation of localization could be achieved using locus zoom plots. I suggest manually checking to eyeball the statistical values reported. If there are truly strong PP4 scores, the patterns of eQTL and pQTL associations OVERALL in the region will appear strongly correlated.

g. To close the loop, one could then also perform formal conditional analysis on the lead SNP. If the sentinel SNP is not the same, but the signals statistically localize, conditional analysis on one (or the other) SNP in the pQTL or eQTL can would ablate each other's signals, respectively. I would perhaps perform this only if one had a really interesting result and want to make sure the signals truly to map to the same signal (and, potentially, the same causal variant). This doesn't need to be done on everything (or anything, if specific pQTL:eQTL pairs aren't interesting enough to warrant it).

3a. Section trans-pQTL and CHD: They authors present a model whereby trans-pQTLs act as they do, because they are cis-eQTLs. I would believe this to the extent that the cis-eQTL and trans pQTL data localize to the same variant.

3b. Moreover, the interpretation of this is challenged somewhat, as the "cis-eQTL" gene that is emphasize for many of the results is PSRC1. For the biology here, it's clear that there is far more compelling data supporting SORT1 as the relevant gene. Because the eQTL for PSRC1 and SORT1 (and another gene in the region) are the same, I think it must be because PSRC1 is an eQTL in blood

and Liver, whereas SORT1 is specific to the liver. I understand why the authors are presenting the data “just the facts” as they are, but they must agree that the biology is more compelling, and that there must be ways to avoid confusion about it — unless they wish to make a case that PSRC1 is another causal gene. (though that would require far more evidence that is presented here).

It could be as simple as presenting the gene tag as “CELSR2/SORT1/PSRC1” in the places where relevant instead of PSRC1. But the tables, main text, etc. do not at all mention any of what’s listed in the rebuttal.

Minor comments

- Abstract:

a. “comprehensively mapping over 16,000 pQTL variants” — this is overstated. The total number of LD-independent sentinel associations is 2-2.5 orders of magnitude smaller than this.

b. “coincided with CAD risk” is not a particularly helpful statement, and again overstates what was actually observed — that is, 1 of the 13 pQTLs statistically co-localized with an established CHD association, and it occurs at a known locus (LPA). “coincided” i.e., ‘coincidental’ association means a lack of mechanistic link, which you formally evaluate

c. the last sentence also seems like a stretch. I appreciate the need from the authors to drum up their work; but given the score of what is in this report, this claim does not feel particularly well justified.

- Figure S3. I felt like I had to stare at this figure for 3 minutes to understand it. This could be visualized in a much better way (with a causal graph, for example).

- I felt like the discussion of genes at the end felt a little much -- some discrepancies certainly are worth talking about, but this all felt a little bit speculative since I wasn't fully convinced by the MR analyses at the end. If presented more as a resource than putting a ton of weight on causal discovery (i.e., calibrated a bit more conservatively), suggesting follow-up efforts that require more work and/or data to proven convincingly, I think this would read much more amicably (at least for my taste, for what that's worth).

Detailed Response

Reviewer #1 (Remarks to the Author):

The authors provide a details response to my concerns. I have additional comments here.

Reply: We thank the reviewer for the useful suggestions for improving our manuscript and have made further revisions in response to each of the reviewer's suggestions as detailed below.

1. Functional, reg, clinical enrichments.

a. In my previous comments, I suggested that because the enrichment analysis based on the proteins pre-selected to related to CHD, any enrichment analysis performed here is to be expected. In looking more closely at this, it doesn't appear that there's any statistical quantification of enrichment at all here (w.r.t. gene enrichment). The use of the specific tool is simply as one for annotation. The annotations w.r.t. gene location are fine. As such, I'm not sure at all what the the last 3 sentences add, or Figure 2. If the genes selected for protein quantification are already CVD related, then all of this is known, enrichments are not tested, so I don't know what any of this actually adds.

b. It looks to me like the KEGG Pathway analysis involves taking all the genes in the interval around a pQTL — not sure this realistically adds anything at all.

c. It also seems like most of the supplement (Figure S2) is a long list of these outputs — again, I'm not sure 100+ pages of these figures really add much and certainly aren't particularly useful.

Reply: We thank the reviewer for these suggestions. We agree that the KEGG pathway analysis did not add much high-value information. Therefore, we removed Figure S2 (100+ pages!). We have retained Figure 2 and the last 3 sentences in the “pQTL Functional and Regulatory Annotations” section. We believe that functional annotation clarifies previously known biological pathways and identifies several novel proteins as candidate CHD drug targets by linking their pQTLs to CHD risk pathways. We have revised the manuscript as follows:

pQTL Functional and Regulatory Annotations: We explored the function annotation of each protein. Some of the genes coding for CHD-related proteins are linked to known CHD risk pathways via previous GWAS of lipids (APOB, LPA, ANGPTL3), coagulation pathways (GMP140), and systemic inflammation (sGP130, sICAM1) as shown in Figure 2. Many of the proteins that share genetic underpinnings with CHD are known drug targets (DrugBank database)¹, or are currently under development as such (e.g. ANGPTL3, LPA, sICAM1, and GMP140). Several proteins with pQTLs linked to CHD, however, are not known drug targets, particularly those from genetic loci not previously linked to CHD risk pathways (e.g. BCHE, CXCL16, MCAM, and sRAGE).

2. Co-localization analysis. The authors respond by performing a colocalization analysis with coloc. The authors state that the first thing they did was “pruned the pQTL variants, retaining only those with LD $r^2 < 0.1$ ”. I'm not entirely sure I understand what this means; the specifics for priors I understand (could be left in methods), but the overall approach as worded is unclear enough to me that I want to make sure this analysis is performed in the way that I expect it to.

Can the author confirm the following general approach:

- a. Identify a locus that harbors a pQTL - e.g., at least one sentinel SNP that is the most strongly associated, surpassing a multiple-test correction. Note the targeted associated gene of interest.
- b. Determine if, for the given target gene, if there exists an associated eQTL for that gene in FHS (blood).
- c. If there exists at least one eQTL associated variant in the interval, perform co-localization, i.e.
- d. Take the regional association data for all variants (say, that spans the territory), for pQTL and eQTL association. This step should involve all SNP associations data, not pruned data (looking at Table S12, not sure what n(SNP) means — this can't be the number of SNPs going into the coloc analysis?)
- e. Report PP1, PP2, PP3, PP4 data. Ideally, $PP4/(PP3+PP4)$ suggests some localization, very high PP4 values denote very strong association
- f. Visual (heuristic) conformation of localization could be achieved using locus zoom plots. I suggest manually checking to eyeball the statistical values reported. If there are truly strong PP4 scores, the patterns of eQTL and pQTL associations OVERALL in the region will appear strongly correlated.
- g. To close the loop, one could then also perform formal conditional analysis on the lead SNP. If the sentinel SNP is not the same, but the signals statistically localize, conditional analysis on one (or the other) SNP in the pQTL or eQTL can would ablate each other's signals, respectively. I would perhaps perform this only if one had a really interesting result and want to make sure the signals truly to map to the same signal (and, potentially, the same causal variant). This doesn't need to be done on everything (or anything, if specific pQTL:eQTL pairs aren't interesting enough to warrant it).

Reply: We thank the reviewer for this excellent suggestion. We have revised the methods used in our colocalization analysis and now use all variants instead of only LD-pruned SNPs.

Methods:

Colocalization of cis-pQTLs and eQTLs: Colocalization analysis involved a two-step procedure. Using our cis-pQTL results, we first identified the locus that harbored the sentinel cis-pQTL variant for each protein. Using FHS eQTL results², we then identified all genes within ± 1 Mb that were also associated with the lead pQTL variant. Because eQTL analysis was based on 1000 Genomes imputed SNPs, we used the lead pQTL variant from 1000 Genomes imputation when the lead pQTL variant was from Exome Chip. Second, because genes at the same locus are often correlated and regulated by the same cis SNPs, we only retained the gene with the lowest SNP-gene association P value for each qualifying sentinel pQTL variant, which resulted in gene-protein pairs showing association with a common SNP and able to be tested for colocalization. To estimate the probability that cis-eQTLs and cis-pQTLs residing in the same genomic location share the same causal variant, we conducted a Bayesian test for colocalization of all eQTL-pQTL pairs using the coloc package in R³. This method requires specifying a prior probability for a SNP being associated with gene expression only (p_1), protein level only (p_2), and with both traits (p_{12}). We applied the default P values, with p_1 and p_2 set to $1E-4$, assuming that 1 in 10,000 SNPs are causal for either trait and p_{12} was set to $1E-5$. We also used p_1 and p_2 values based on the number of eQTL and pQTL variants observed in our data. For the eQTL analysis, we detected 19,613 non-redundant eSNPs among 4,285,456 total cis-SNPs, indicating that the probability of a

SNP being a causal eSNP is $4.6E-3$. This probability corresponds to the sum $p_1 + p_{12}$. For the pQTL analysis, we detected 254 non-redundant cis-pQTL variants among 440,409 total cis-SNPs, indicating that the probability that a SNP is a causal pQTL variant is $5.7E-4$. This probability corresponds to the sum $p_2 + p_{12}$. p_{12} was set to 0.003 corresponding to a probability of 75% that a causal eSNP is also a causal pQTL variant, an approach that has been shown to represent the best choice⁴.”

Results:

Colocalization of pQTLs and eQTLs: Among the 372 non-redundant pQTL variants, we identified 190 unique variants (associated with 53 proteins) that were also eQTL variants (genetic variants associated with whole blood gene expression levels in FHS participants²) at $FDR < 0.05$. These 190 eQTL variants consisted of 188 cis-eQTL variants and 27 trans-eQTL variants (Table S11), suggesting that a substantial number of pQTL variants may share causal variants with eQTLs. To test this hypothesis, we conducted a Bayesian test of colocalization of cis-pQTL variants using the coloc package in R³ (see Methods). Among the 40 sentinel cis-pQTL variants, 26 were associated with the expression of genes residing within 1 Mb ($FDR < 0.05$), and these 26 unique lead SNP-transcript-protein pairs to test for colocalization. Using all SNPs shared by transcripts and proteins, we conducted a colocalization test for each protein to determine the probability that the two association signals were due to the same causal variant. The prior probabilities for a SNP being associated with gene expression only (p_1), protein level only (p_2), or with both traits (p_{12}) were based on the number of eSNPs and pQTL variants observed in our data (see Methods). The value for p_{12} was set to 75%, i.e. the probability that a causal eSNP is a causal pQTL variant. For 16 out of 26 proteins that were associated with both cis-pQTL variants and eQTL variants, we observed a probability $> 75\%$ that the pQTL variants colocalized with the eQTL variants (Table S12). We observed similar colocalization results by applying default P values ($p_1 = 1E-4$, $p_2 = 1E-4$, $p_{12} = 1E-5$) in the coloc package assuming that 1 in 10,000 SNPs is causal for either trait.

Table S12. Proteins with $> 75\%$ probability of colocalization of pQTL variants with eQTLs variants (PP.H4 value > 0.75 represents colocalization of pQTLs and eQTLs)

Protein	Target gene	SNPs	PP.H0	PP.H1	PP.H2	PP.H3	PP.H4
CXCL16	CXCL16	1	1.06E-76	3.82E-68	9.39E-13	0	1
CD14	CD14	223	3.18E-80	8.72E-35	5.49E-49	0.001	0.999
MCAM	CBL	6	4.31E-09	3.09E-06	2.23E-06	0.001	0.999
MMP9	MMP9	9	3.70E-18	0.0002646	2.44E-17	0.001	0.998
MMP8	MMP8	1	2.07E-16	0.0050028	1.39E-17	0	0.995
GRN	HIGD1B	55	3.97E-17	3.41E-05	6.91E-15	0.006	0.994
GDF15	PGPEP1	18	6.16E-70	5.09E-05	1.17E-67	0.009	0.991
FGG	PLRG1	49	5.05E-16	9.30E-09	5.59E-10	0.00995	0.990
sGP130	IL6ST	63	1.17E-32	3.51E-23	4.39E-12	0.0128	0.987
A1M	ORM1	3	1.93E-35	4.62E-31	7.89E-07	0.0185	0.981
COL18A1	COL18A1	106	2.77E-126	3.39E-117	2.41E-11	0.029	0.971

AGP1	ORM1	7	6.93E-52	5.87E-44	5.02E-10	0.042	0.958
sRAGE	PBX2	6	7.95E-49	0.0024945	3.05E-47	0.095	0.902
Cystatin C	CST3	116	4.29E-71	1.62E-05	4.01E-67	0.152	0.848
MPO	SUPT4H1	567	2.48E-149	5.98E-138	7.97E-13	0.192	0.808
ADM	ADM	419	1.89E-28	1.54E-17	2.70E-12	0.220	0.780
GP5	ATP13A3	89	4.87E-30	2.11E-19	8.12E-12	0.351	0.649
UCMGP	MGP	262	3.51E-61	7.61E-23	4.62E-39	0.999	5.89E-04
sICAM1	ICAM3	9	8.25E-47	4.07E-34	2.03E-13	1.000	1.41E-05
CLEC3B	ZNF502	26	8.87E-79	6.69E-61	1.33E-18	1	1.11E-10
C2	NCR3	178	1.60E-66	6.51E-41	2.45E-26	1	4.39E-17
SAA1	HPS5	105	5.15E-154	1.80E-26	2.86E-128	1	1.61E-17
NTproBNP	MFN2	147	1.54E-75	2.74E-41	5.61E-35	1	1.59E-24
CD5L	FCRL3	429	1.31E-307	5.11E-258	2.56E-50	1	8.76E-32
GMP140	F5	165	5.27E-277	8.14E-196	6.47E-82	1	2.11E-70
SERPINA10	SERPINA1	250	0	4.22E-124	0	1	6.22E-120

PP.H0 (no causal variant), PP.H1 (causal variant for gene expression only), PP.H2 (causal variant for protein only), PP.H3 (two distinct causal variants), PP.H4 (one common causal variant)

3a. Section trans-pQTL and CHD: They authors present a model whereby trans-pQTLs act as they do, because they are cis-eQTLs. I would believe this to the extent that the cis-eQTL and trans pQTL data localize to the same variant.

3b. Moreover, the interpretation of this is challenged somewhat, as the “cis-eQTL” gene that is emphasize for many of the results is PSRC1. For the biology here, it’s clear that there is far more compelling data supporting SORT1 as the relevant gene. Because the eQTL for PSRC1 and SORT1 (and another gene in the region) are the same, I think it must be because PSRC1 is an eQTL in blood and Liver, whereas SORT1 is specific to the liver. I understand why the authors are presenting the data “just the facts” as they are, but they must agree that the biology is more compelling, and that there must be ways to avoid confusion about it — unless they wish to make a case that PSRC1 is another causal gene. (though that would require far more evidence that is presented here). It could be as simply as presenting the gene tag as “CELSR2/SORT1/PSRC1” in the places where relevant instead of PSRC1. But the tables, main text, etc. do not at all mention any of what’s listed in the rebuttal.

Reply: We thank the reviewer for this suggestion. We have replaced *PSRC1* with *CELSR2/SORT1/PSRC1* in the manuscript and supplementary tables as follows:

Results:

Moreover, we found significant causal effects of CELSR2/SORT1/PSRC1 on APOB levels (in liver), CELSR2/SORT1/PSRC1 on GRN levels (in artery), and ABO on GMP140 levels (in heart atrial appendage).

Table S14: Mendelian randomization analysis of cis-eGenes as regulators of protein level

cis-eGene locus	Proteins	Instrumental variable (trans-pQTL SNPs that are eQTLs)	Beta	SE	P value	Data
CELSR2/SORT1/PSRC1	APOB	rs12740374	-9.541	1.176	4.95E-16	FHS whole blood
ATXN2/SH2B3	B2M	rs10774625	-16.836	2.595	8.66E-11	FHS whole blood
HNF1A/P2RX4	CRP	rs7979473	-10.775	1.638	4.74E-11	FHS whole blood
ABO	GMP140	rs2519093	17.350	0.923	7.55E-79	FHS whole blood
CELSR2/SORT1/PSRC1	GRN	2 SNPs	-39.325	18.682	3.53E-02	FHS whole blood
ABO	MCAM	rs550057	5.959	0.868	6.71E-12	FHS whole blood
ABO	sGP130	rs532436	8.316	0.932	4.66E-19	FHS whole blood
ABO	sICAM1	rs507666	12.054	0.887	4.39E-42	FHS whole blood
CELSR2/SORT1/PSRC1	APOB	rs629301	-0.650	0.080	2.45E-15	GTEEx Whole blood
CELSR2/SORT1/PSRC2	GRN	rs629301	-2.700	0.080	4.83E-264	GTEEx Whole blood
CELSR2/SORT1/PSRC3	APOB	rs7528419	-0.135	0.017	3.53E-15	GTEEX Liver
CELSR2/SORT1/PSRC4	GRN	rs35558471	-0.567	0.105	7.50E-08	GTEEx Artery Aorta
ABO	GMP140	rs11244053	-0.133	0.024	1.71E-08	GTEEx Heart Atrial Appendage

Minor comments-

Abstract:

- a. “comprehensively mapping over 16,000 pQTL variants” — this is overstated. The total number of LD-independent sentinel associations is 2-2.5 orders of magnitude smaller than this.
- b. “coincided with CAD risk” is not a particularly helpful statement, and again overstates what was actually observed — that is, 1 of the 13 pQTLs statistically co-localized with an established CHD

association, and it occurs at a known locus (LPA). “coincided” i.e., ‘coincidental’ association means a lack of mechanistic link, which you formally evaluate

c. the last sentence also seems like a stretch. I appreciate the need from the authors to drum up their work; but given the score of what is in this report, this claim does not feel particularly well justified.

Reply: We thank the reviewer for these suggestions. We have revised the abstract as follows:

Identifying genetic variants associated with circulating protein concentrations (protein quantitative trait loci; pQTLs) and integrating them with variants from genome-wide association studies (GWAS) may illuminate the proteome’s causal role in disease and bridge a GWAS knowledge gap regarding unexplained SNP-disease associations. We conducted GWAS of 71 high-value proteins for cardiovascular disease in 6,861 Framingham Heart Study participants and then externally replicated our findings. We comprehensively mapped over 16,000 pQTL variants (372 non-redundant), explored their functional relevance, and created an integrated plasma-protein-QTL database. Next, we identified 13 proteins with pQTLs that either matched coronary heart disease-risk variants from GWAS or tested causal for coronary disease by Mendelian randomization. Eight of these proteins were predictive of new-onset cardiovascular disease events in Framingham Heart Study participants with long-term follow-up. This body of work demonstrates that identifying pQTLs, integrating them with GWAS results, employing Mendelian randomization, and testing protein-trait associations holds the potential for elucidating genes, proteins, and pathways that are causally associated with cardiovascular disease and may identify novel therapeutic targets for its treatment and prevention.

- Figure S3. I felt like I had to stare at this figure for 3 minutes to understand it. This could be visualized in a much better way (with a causal graph, for example).

Reply: We agree with the reviewer. We have revised the Results and figure S3 as follows:

Results:

Trans-pQTLs and CHD: Of the nine proteins with pQTL variants that precisely matched CHD-associated SNPs, eight had pQTLs with trans effects (54 non-redundant trans-pQTL variants in total) and 69% of these trans-pQTL variants were also associated with the expression of nearby genes (cis-eGenes), i.e. these trans-pQTL variants were also cis-eQTL variants associated with the expression of nearby cis-eGenes. Based on these findings, we hypothesized that trans-pQTL variants may regulate circulating protein levels through cis-effects on the expression of nearby cis-eGenes (Figure S2a). To test this hypothesis, we employed Mendelian randomization (MR)⁵ using the expression of all genes within 1 Mb of the trans-pQTL locus as the exposure, cis-eQTLs associated with these genes (from the FHS whole blood gene expression database²) as instrumental variables, and circulating protein levels as the outcome. We found that for eight proteins the effects of trans-pQTLs on circulating protein levels were causally regulated by the expression of cis-eGenes (Table S14). For example, we found decreased SH2B3 expression to be causal for higher circulating B2M levels. This extends prior knowledge of the associations of

*SH2B3*⁶ and *B2M*⁷ with hypertension as we demonstrate a unidirectional causal association between *SH2B3* expression and plasma *B2M* levels, and thus provide plausible evidence for a causal role of the *SH2B3*-*B2M* axis in hypertension (Figure S2b). To extend these findings to other CHD-related tissues, we applied the same analyses to GTEx⁸ whole blood, liver, and heart eQTLs. For two of the proteins (*APOB* and *GRN*), there was additional experimental evidence in support of our results through interrogation of GTEx⁹ whole blood eQTLs (Table S14). Moreover, we found significant causal effects of *CELSR2/SORT1/PSRC1* on *APOB* levels (in liver), *CELSR2/SORT1/PSRC1* on *GRN* levels (in artery), and *ABO* on *GMP140* levels (in heart atrial appendage).

Figure S2. *Trans*-pQTL variants regulate circulating protein levels through the expression of nearby *cis*-eGenes

Figure 2a

Figure 2a: *Trans*-pQTL variants regulate circulating protein levels through the expression of nearby *cis*-eGenes: a pQTL variant (SNP) is a *cis*-eQTL for Gene A (*Cis*-eGene) and a *trans*-pQTL for Protein B (Protein). Mendelian randomization establishes a causal effect of Gene A expression on circulating Protein B levels.

Figure 2b

Figure 2b: Example of circulating B2M levels regulated by *SH2B3* expression. The A allele of rs10774625 (*ATXN2/SH2B3* locus) reduces *SH2B3* expression, which in turn increases CD8+ T cell differentiation⁶ and circulating B2M levels¹⁰. The resulting increases in CD8+ T cells and plasma B2M levels result in a greater risk of hypertension^{7,11}.

- I felt like the discussion of genes at the end felt a little much -- some discrepancies certainly are worth talking about, but this all felt a little bit speculative since I wasn't fully convinced by the MR analyses at the end. If presented more as a resource than putting a ton of weight on causal discovery (i.e., calibrated a bit more conservatively), suggesting follow-up efforts that require more work and/or data to proven convincingly, I think this would read much more amicably (at least for my taste, for what that's worth).

Reply: We thank the reviewer for this suggestion, and have condensed the section on "Novel Proteins and Pathways Implicated in CHD" as follows:

Novel Proteins and Pathways Implicated in CHD: Six proteins were implicated by MR analyses as nominally causal for CHD (Table S15). Lipoprotein (a) (LPA), which interferes with the fibrinolytic cascade¹², has been previously demonstrated to be causal for CHD¹³, and served as a positive control. Butyrylcholinesterase (BCHE) has previously been reported to be inversely associated

with long-term CVD mortality^{14,15}, and several polymorphisms within BCHE have been reported¹⁶ to be associated with CHD risk factors. rs1803274, the sentinel cis-pQTL variant for BCHE (A539T) (Table S6), is associated with decreased BCHE circulating levels and enzymatic activity¹⁷, and has been shown to predict early-onset CHD¹⁶. Our protein-trait analyses similarly demonstrated inverse associations between plasma BCHE and long-term cardiovascular outcomes, and MR analyses revealed lower BCHE to be causal for CHD.

Four additional proteins were nominally causal for CHD by MR. PON1 exhibits cardioprotective effects through prevention of LDL oxidation¹⁸, and overexpression of PON1 in mice inhibits the development of atherosclerosis¹⁹. Our protein-trait analyses similarly demonstrated an inverse association between PON1 levels and long-term cardiovascular outcomes, but MR revealed that higher PON1 levels are causal for CHD (Table 2). We hypothesize that this directional discordance may reflect down-regulation of PON1 expression in the setting of CHD. Myeloperoxidase (MPO), which promotes formation of atherosclerotic lesions by enhancing apolipoprotein B (APOB) oxidation within circulating LDL particles²⁰, was positively associated with incident cardiovascular outcomes in our protein-trait and MR analyses. Cystatin C, a pro-atherosclerotic²¹ cysteine proteinase cathepsin inhibitor and well-characterized biomarker of CHD risk²², was also positively associated with CVD events in our protein-trait and MR analyses. MCAM, or CD146, was causally associated with CHD risk in an inverse manner by MR. This is directionally consistent with prior animal studies of limb ischemia, which have shown that injection of sCD146 into the circulation decreased fibrosis and inflammation and increased local perfusion²³.

Several proteins lacked cis-pQTLs and therefore were unavailable for MR analysis. Six of these proteins, which had pQTL variants that perfectly matched CHD SNPs from GWAS, were associated with CHD/CVD outcomes in FHS participants with long-term follow-up: GRN, sGP130, sICAM1, APOB, B2M, and CRP. GRN has previously been implicated in atherosclerosis progression and incident MI^{24,25}. Its precursor, progranulin, has been shown to bind to SORT1, which contains a trans-pQTL variant for GRN that is also associated with CHD²⁶. sGP130 levels have been shown to positively correlate with long-term CVD mortality²⁷, perhaps via pathways related to hypertension and vascular remodeling²⁸. B2M, an essential component of the major histocompatibility complex I²⁹, is associated with hypertension³⁰, atherosclerosis, and CVD³¹. Finally, circulating APOB, an LDL particle ligand, is a well-characterized biomarker of CVD risk.^{32,33}

References:

- 1 Law, V. et al. DrugBank 4.0: shedding new light on drug metabolism. *Nucleic Acids Res* **42**, D1091-1097, doi:10.1093/nar/gkt1068 (2014).
- 2 Joehanes, R. et al. Integrated genome-wide analysis of expression quantitative trait loci aids interpretation of genomic association studies. *Genome biology* **18**, 16, doi:10.1186/s13059-016-1142-6 (2017).

- 3 Giambartolomei, C. *et al.* Bayesian test for colocalisation between pairs of genetic association studies using summary statistics. *PLoS Genet* **10**, e1004383, doi:10.1371/journal.pgen.1004383 (2014).
- 4 Pierce, B. L. *et al.* Co-occurring expression and methylation QTLs allow detection of common causal variants and shared biological mechanisms. *Nat Commun* **9**, 804, doi:10.1038/s41467-018-03209-9 (2018).
- 5 Smith, G. D. & Ebrahim, S. 'Mendelian randomization': can genetic epidemiology contribute to understanding environmental determinants of disease? *Int J Epidemiol* **32**, 1-22 (2003).
- 6 Dale, B. L. & Madhur, M. S. Linking inflammation and hypertension via LNK/SH2B3. *Curr Opin Nephrol Hypertens* **25**, 87-93, doi:10.1097/MNH.000000000000196 (2016).
- 7 You, L. *et al.* High levels of serum beta2-microglobulin predict severity of coronary artery disease. *BMC Cardiovasc Disord* **17**, 71, doi:10.1186/s12872-017-0502-9 (2017).
- 8 Consortium, G. T. The Genotype-Tissue Expression (GTEx) project. *Nat Genet* **45**, 580-585, doi:10.1038/ng.2653 (2013).
- 9 Consortium, G. T. Human genomics. The Genotype-Tissue Expression (GTEx) pilot analysis: multitissue gene regulation in humans. *Science* **348**, 648-660, doi:10.1126/science.1262110 (2015).
- 10 Huan, T. *et al.* Integrative network analysis reveals molecular mechanisms of blood pressure regulation. *Molecular systems biology* **11**, 799, doi:10.15252/msb.20145399 (2015).
- 11 Itani, H. A. *et al.* Activation of Human T Cells in Hypertension: Studies of Humanized Mice and Hypertensive Humans. *Hypertension* **68**, 123-132, doi:10.1161/HYPERTENSIONAHA.116.07237 (2016).
- 12 Maranhao, R. C., Carvalho, P. O., Strunz, C. C. & Pileggi, F. Lipoprotein (a): structure, pathophysiology and clinical implications. *Arq Bras Cardiol* **103**, 76-84 (2014).
- 13 Zhao, W., Lee, J. J., Rasheed, A. & Saleheen, D. Using Mendelian Randomization studies to Assess Causality and Identify New Therapeutic Targets in Cardiovascular Medicine. *Curr Genet Med Rep* **4**, 207-212, doi:10.1007/s40142-016-0103-4 (2016).
- 14 Santarpia, L., Grandone, I., Contaldo, F. & Pasanisi, F. Butyrylcholinesterase as a prognostic marker: a review of the literature. *J Cachexia Sarcopenia Muscle* **4**, 31-39, doi:10.1007/s13539-012-0083-5 (2013).
- 15 Calderon-Margalit, R., Adler, B., Abramson, J. H., Gofin, J. & Kark, J. D. Butyrylcholinesterase activity, cardiovascular risk factors, and mortality in middle-aged and elderly men and women in Jerusalem. *Clin Chem* **52**, 845-852, doi:10.1373/clinchem.2005.059857 (2006).
- 16 Nassar, B. A. *et al.* Relation between butyrylcholinesterase K variant, paraoxonase 1 (PON1) Q and R and apolipoprotein E epsilon 4 genes in early-onset coronary artery disease. *Clin Biochem* **35**, 205-209 (2002).
- 17 The UniProt Consortium. UniProt: the universal protein knowledgebase. *Nucleic Acids Research* **45**, D158-D169, doi:10.1093/nar/gkw1099 (2017).
- 18 Cheraghi, M., Shahsavari, G., Maleki, A. & Ahmadvand, H. Paraoxonase 1 Activity, Lipid Profile, and Atherogenic Indexes Status in Coronary Heart Disease. *Rep Biochem Mol Biol* **6**, 1-7 (2017).
- 19 Mackness, B., Quarck, R., Verreth, W., Mackness, M. & Holvoet, P. Human paraoxonase-1 overexpression inhibits atherosclerosis in a mouse model of metabolic syndrome. *Arteriosclerosis, thrombosis, and vascular biology* **26**, 1545-1550, doi:10.1161/01.ATV.0000222924.62641.aa (2006).
- 20 Teng, N. *et al.* The roles of myeloperoxidase in coronary artery disease and its potential implication in plaque rupture. *Redox Rep* **22**, 51-73, doi:10.1080/13510002.2016.1256119 (2017).

- 21 Wu, H., Du, Q., Dai, Q., Ge, J. & Cheng, X. Cysteine Protease Cathepsins in Atherosclerotic Cardiovascular Diseases. *J Atheroscler Thromb*, doi:10.5551/jat.RV17016 (2017).
- 22 Gu, F. F. *et al.* Relationship between plasma cathepsin S and cystatin C levels and coronary plaque morphology of mild to moderate lesions: an in vivo study using intravascular ultrasound. *Chin Med J (Engl)* **122**, 2820-2826 (2009).
- 23 Harhour, K. *et al.* Soluble CD146 displays angiogenic properties and promotes neovascularization in experimental hind-limb ischemia. *Blood* **115**, 3843-3851, doi:10.1182/blood-2009-06-229591 (2010).
- 24 Yin, X. *et al.* Protein biomarkers of new-onset cardiovascular disease: prospective study from the systems approach to biomarker research in cardiovascular disease initiative. *Arteriosclerosis, thrombosis, and vascular biology* **34**, 939-945, doi:10.1161/atvbaha.113.302918 (2014).
- 25 Kojima, Y. *et al.* Progranulin expression in advanced human atherosclerotic plaque. *Atherosclerosis* **206**, 102-108, doi:10.1016/j.atherosclerosis.2009.02.017 (2009).
- 26 Hu, F. *et al.* Sortilin-mediated endocytosis determines levels of the frontotemporal dementia protein, progranulin. *Neuron* **68**, 654-667, doi:10.1016/j.neuron.2010.09.034 (2010).
- 27 Askevold, E. T. *et al.* Soluble glycoprotein 130 predicts fatal outcomes in chronic heart failure: analysis from the Controlled Rosuvastatin Multinational Trial in Heart Failure (CORONA). *Circ Heart Fail* **6**, 91-98, doi:10.1161/CIRCHEARTFAILURE.112.972653 (2013).
- 28 Morieri, M. L., Passaro, A. & Zuliani, G. Interleukin-6 "Trans-Signaling" and Ischemic Vascular Disease: The Important Role of Soluble gp130. *Mediators Inflamm* **2017**, 1396398, doi:10.1155/2017/1396398 (2017).
- 29 Argyropoulos, C. P. *et al.* Rediscovering Beta-2 Microglobulin As a Biomarker across the Spectrum of Kidney Diseases. *Front Med (Lausanne)* **4**, 73, doi:10.3389/fmed.2017.00073 (2017).
- 30 Huang, M. *et al.* Association of kidney function and albuminuria with prevalent and incident hypertension: the Atherosclerosis Risk in Communities (ARIC) study. *Am J Kidney Dis* **65**, 58-66, doi:10.1053/j.ajkd.2014.06.025 (2015).
- 31 Wu, H. C., Lee, L. C. & Wang, W. J. Associations among Serum Beta 2 Microglobulin, Malnutrition, Inflammation, and Advanced Cardiovascular Event in Patients with Chronic Kidney Disease. *J Clin Lab Anal* **31**, doi:10.1002/jcla.22056 (2017).
- 32 Boekholdt, S. M. *et al.* Association of LDL cholesterol, non-HDL cholesterol, and apolipoprotein B levels with risk of cardiovascular events among patients treated with statins: a meta-analysis. *JAMA* **307**, 1302-1309, doi:10.1001/jama.2012.366 (2012).
- 33 Sniderman, A. D. *et al.* A meta-analysis of low-density lipoprotein cholesterol, non-high-density lipoprotein cholesterol, and apolipoprotein B as markers of cardiovascular risk. *Circ Cardiovasc Qual Outcomes* **4**, 337-345, doi:10.1161/CIRCOUTCOMES.110.959247 (2011).

Reviewer #1 (Remarks to the Author):

The authors provide a second, detailed response to my concerns.

I'm still a little bit confused on Co-Localization.

1. I apologize for being daft, but I still confused as to what is present in the column labeled "SNPs" in Table S12.

My understanding for co-localization is that one should take an interval around each respective signal - and one that is common for both traits.

I would have thought this would have been somewhat straight forward — around the eQTL/pQTL locus, define a physical region which captures the association signal(s) for both traits

Then, perform `coloc()`.

It seem like in methods that the authors are reporting using all SNPs in the region — so I am trusting that this is in order.

But there isn't much detail defining regional 'span' around the association signals; add that to the "SNPs" column, and that leaves me a little confused.

Perhaps they just need to clarify what's in that column, explain in methods how the physical intervals from which `coloc()` is perform are defined, and that will be sufficient.

2. The authors also present a subtly that I did not fully appreciate - though I suppose could happen: a cis-pQTL and cis-eQTL could have localization, but to different genes.

looking at S12, it seems to me that there's an pretty strong enrichment for High PP3 Probs and Protein:Target eQTL MISmapping,

i.e., of the 9 analyses with $PP3 \approx 1.0$, all of them are examples where Protein:target gene are different.

This easily makes sense: You could have an eQTL for transcript X that has nothing to do with the pQTL, which is for transcript Y, and both are strongly associated (but not the same variant).

That's why in the original comment, I suggested that the authors focus on cases where the eQTL and pQTL both mapped to the same "Gene ID" (protein and target gene are the same).

You can see that of those with $PP4 > 0.75$, 6 of 16 are the same. But that seems lower than what I might have thought.

I'm reading and it seems like the authors based their co-loc for eQTLs on perhaps the strongest eQTL in the region

that's perhaps fine, but I think in my mind, the most important thing would be "synergy" across eQTL/pQTL gene target.

I will concede that this can be complex (the absence of an eQTL evidence does not mean there is no eQTLs for the target gene).

However, the pQTL was performed in a rather general way — so I would have expected eQTLs to be seen.

Variation that obviously impacts gene expression may easily have an impact on protein abundance.

Detailed Response

Reply: We thank the reviewer for these comments. We have revised the text to clarify the co-localization section.

1. I apologize for being daft, but I still confused as to what is present in the column labeled “SNPs” in Table S12. My understanding for co-localization is that one should take an interval around each respective signal - and one that is common for both traits. I would have thought this would have been somewhat straight forward — around the eQTL/pQTL locus, define a physical region which captures the association signal(s) for both traits. Then, perform coloc(). It seem like in methods that the authors are reporting using all SNPs in the region — so I am trusting that this is in order. But there isn’t much detail defining regional ‘span’ around the association signals; add that to the “SNPs” column, and that leaves me a little confused. Perhaps they just need to clarify what’s in that column, explain in methods how the physical intervals from which coloc() is perform are defined, and that will be sufficient.

Reply: We defined the colocalization region in the Methods section as follows: “Using FHS eQTL results, we identified all genes within ± 1 Mb of an eQTL that were also associated with the lead pQTL variant.” To avoid confusion about the region used for colocalization testing, we modified the Methods and Results sections as follows:

Methods:

“Colocalization analysis involved a two-step procedure. Using our cis-pQTL results, we first identified the locus that harbored the sentinel cis-pQTL variant for each protein. The locus was defined as a 1 Mb region (upstream and downstream) from each sentinel cis-pQTL variant. Using FHS eQTL results, we then identified all genes within each locus for which expression of a gene was associated with the lead pQTL variant.”

Results:

“To test this hypothesis, we conducted a Bayesian test of colocalization of cis-pQTL variants using the coloc package in R for genes within 1 Mb (upstream or downstream) of each sentinel cis-pQTL variant (see Methods).”

2. The authors also present a subtlety that I did not fully appreciate - though I suppose could happen: a cis-pQTL and cis-eQTL could have localization, but to different genes. looking at S12, it seems to me that there’s an pretty strong enrichment for High PP3 Probs and Protein:Target eQTL MISmapping, i.e., of the 9 analyses with PP3 ~ 1.0 , all of them are examples where Protein:target gene are different. This easily makes sense: You could have an eQTL for transcript X that has nothing to do with the pQTL, which is for transcript Y, and both are strongly associated (but not the same variant). That’s why in the original comment, I suggested that the authors focus on cases where the eQTL and pQTL both mapped to the same “Gene ID” (protein and target gene are the same). You can see that of those with PP4 > 0.75, 6 of 16 are the same. But that seems lower than what I might have

thought. I'm reading and it seems like the authors based their co-loc for eQTLs on perhaps the strongest eQTL in the region that's perhaps fine, but I think in my mind, the most important thing would be "synergy" across eQTL/pQTL gene target. I will concede that this can be complex (the absence of an eQTL evidence does not mean there is no eQTLs for the target gene). However, the pQTL was performed in a rather general way — so I would have expected eQTLs to be seen. Variation that obviously impacts gene expression may easily have an impact on protein abundance.

Reply: The reviewer is correct. Our co-localization analysis approach is based on the strongest eQTL in the region (1 Mb upstream or downstream of a sentinel pQTL), which is the same method used in the seminal co-localization paper that we cited: "Pierce, B. L. et al. Co-occurring expression and methylation QTLs allow detection of common causal variants and shared biological mechanisms. Nat Commun 9, 804, doi:10.1038/s41467-018-03209-9 (2018)." The purpose of the approach is to generate one protein-gene pair within each region, which will decrease the number of false positive results by using all genes within the 1 Mb region. We understand that the reviewer expected to see a larger proportion of protein coding genes that co-localize with their corresponding proteins, which is the ideal situation. However, because gene expression and circulating protein levels vary in time and we did not measure protein levels and gene expression at the same time point, it is not surprising that protein and target gene have two distinct causal variants (high PP3). We modified the discussion as follows:

"We acknowledge several limitations of our study...Finally, protein levels were measured in whole blood and may not accurately reflect tissue-specific patterns of expression. Furthermore, our gene transcript levels and circulating protein levels were not measured at the same point in time, which may limit the power to find colocalization of protein coding genes with their corresponding proteins."